# Regulation of human trophoblast gene expression by endogenous retroviruses

Jennifer M. Frost [1]✉, Samuele M. Amante[1], Hiroaki Okae[2], Eleri M. Jones[1], Brogan Ashley[3], Rohan M. Lewis[3], Jane K. Cleal[3], Matthew P. Caley[1], Takahiro Arima[4], Tania Maffucci[1] & Miguel R. Branco[1]✉

The placenta is a fast-evolving organ with large morphological and histological differences across eutherians, but the genetic changes driving placental evolution have not been fully elucidated. Transposable elements, through their capacity to quickly generate genetic variation and affect host gene regulation, may have helped to define species-specific trophoblast gene expression programs. Here we assess the contribution of transposable elements to human trophoblast gene expression as enhancers or promoters. Using epigenomic data from primary human trophoblast and trophoblast stem-cell lines, we identified multiple endogenous retrovirus families with regulatory potential that lie close to genes with preferential expression in trophoblast. These largely primate-specific elements are associated with inter-species gene expression differences and are bound by transcription factors with key roles in placental development. Using genetic editing, we demonstrate that several elements act as transcriptional enhancers of important placental genes, such as *CSF1R* and *PSG5*. We also identify an LTR10A element that regulates *ENG* expression, affecting secretion of soluble endoglin, with potential implications for preeclampsia. Our data show that transposons have made important contributions to human trophoblast gene regulation, and suggest that their activity may affect pregnancy outcomes.

The success of human pregnancy depends on the healthy development and function of the placenta. Following implantation, fetally derived trophoblast cells invade maternal tissues interstitially, remodeling uterine spiral arteries well into the myometrium[1]. Aberrations to this process result in serious pregnancy complications that cause maternal and fetal morbidity and mortality, including recurrent pregnancy loss, fetal growth restriction, preterm birth and preeclampsia in the case of too little invasion, or disorders of the placenta accreta spectrum where invasion is too extensive[2]. However, the genetic determinants of these disorders remain unclear, as genome-wide association studies

have revealed very few candidates, with the notable exception of *FLT1* in preeclampsia[3].

Placental development and structure displays wide variation across eutherian species, even within the primate order[4]. Notably, the deep interstitial trophoblast invasion observed in humans is unique to great apes[5]. Placentas differ in cellular composition, histological arrangement and gross morphology, as well as many molecular aspects, all of which shape interactions between conceptus and mother, and their outcomes. This striking variation reflects the myriad selective pressures associated with the feto-maternal conflicts that fuel fast

[1]Blizard Institute, Faculty of Medicine and Dentistry, Queen Mary University of London, London, UK. [2]Department of Trophoblast Research, Institute of Molecular Embryology and Genetics, Kumamoto University, Kumamoto, Japan. [3]School of Human Development and Health, Faculty of Medicine, University of Southampton, Southampton, UK. [4]Department of Informative Genetics, Environment and Genome Research Center, Tohoku University Graduate School of Medicine, Sendai, Japan. ✉e-mail: j.frost@qmul.ac.uk; m.branco@qmul.ac.uk

evolution of this organ[6]. Yet, the genetic drivers of placental evolution remain to be fully elucidated.

Transposable elements (TEs) are one important yet understudied source of genetic variation. These abundant repetitive elements, which include endogenous retroviruses (ERVs), have made major contributions to human evolution, helping to shape both the coding and regulatory (noncoding) landscape of the genome. Akin to the variable and species-specific development and structure of the placenta, TEs are highly species-specific, making them putative drivers of placental evolution. Indeed, multiple genes with key roles in placentation have been derived from TEs[7], most prominently the syncytin genes, whose products mediate cell–cell fusion to generate a syncytialized trophoblast layer that directly contacts the maternal blood[8]. Additionally, the noncoding portions of TEs (for example, the long terminal repeats (LTRs) in ERVs) have the ability to regulate gene transcription, and through this action contribute to human embryonic development[9–11], innate immunity[12], the development of cancer[13] and the evolution of the feto-maternal interface, among others[14,15]. TEs can recruit host transcription factors, often in a highly tissue-specific manner, and gain epigenetic hallmarks of gene regulatory activity, acting as transcriptional promoters or distal enhancer elements[14–16]. We and others have previously shown that in mouse trophoblast stem cells, several ERV families can act as major enhancers of gene expression[17,18]. In humans, several examples of TE-encoded placenta-specific promoters have been uncovered, such as those driving expression of *CYP19A1*, *NOS3* and *PTN* genes[19]. A fascinating example of a human placental TE-derived enhancer has also been described that affects gestational length when inserted into the mouse genome[20]. More recently, the Macfarlan laboratory has identified a group of putative lineage-specific placental enhancers that are derived from ERVs[21]. However, as the placenta is a heterogeneous tissue, it remains unclear whether all of these ERVs are active in trophoblast cells, and a genetic demonstration of their regulatory action is lacking.

In this Article, we identify ERV families that exhibit hallmarks of gene regulatory activity in human trophoblast. We show that these ERVs bind transcription factors required for placental development and lie close to genes with preferential trophoblast expression in a species-specific manner. Using genetic editing, we show examples of ERVs that act as gene enhancers in trophoblast, including an LTR10A element within the *ENG* gene that regulates the secretion of soluble ENG protein by the syncytiotrophoblast (SynT), which is both a marker for, and contributor to, the pathogenesis of preeclampsia.

## Results

### ERVs with regulatory potential in human trophoblast

To identify interspersed repetitive elements bearing hallmarks of activating regulatory potential in human trophoblast, we performed H3K27ac profiling using either chromatin immunoprecipitation sequencing (ChIP-seq) or CUT&Tag (cleavage under targets and tagmentation)[22]. We analyzed our previously published data from primary human cytotrophoblast[23], as well as newly generated data from cytotrophoblast-like human trophoblast stem cells (hTSCs)[24], which can be differentiated in vitro and allow for easy genetic manipulation (Fig. 1a). Using the RepeatMasker annotation, we determined the frequency of H3K27ac peaks per repeat family and compared it with random controls using a permutation test (Fig. 1a). This revealed 29 repeat families enriched for H3K27ac peaks in both primary cytotrophoblast and hTSCs, the vast majority of which were primate-specific ERVs (Fig. 1b, Extended Data Fig. 1a and Supplementary Table 1). For comparison, we performed the same analysis on published H3K27ac CUT&Tag data from human embryonic stem cells (hESCs)[22]. Most hTSC-enriched repeats displayed little to no enrichment in hESCs (Fig. 1b), despite the fact that hESCs make use of a large set of TEs for gene-regulatory purposes[10,25]. A more detailed analysis of H3K27ac signals across each of the enriched families confirmed the asymmetry between hTSCs

and hESCs (Fig. 1c). Human ERV subfamily H (HERVH)-associated LTRs present an interesting case, with LTR7C elements H3K27ac-enriched in hTSCs, whereas the related LTR7 family is H3K27ac-enriched in hESCs (Extended Data Fig. 1b). Each of the LTR7 subfamilies displays a unique combination of transcription-factor-binding motifs[26], which may underlie their cell-type-specific expression.

To further validate our findings and narrow down the list of candidate regulatory repeat families, we also analyzed placental DNAse-seq data from ENCODE. When compared with the data from liver, lung and kidney, most hTSC-associated families displayed tissue-specific enrichment of elements with accessible chromatin (Fig. 1d; non-tissue-specific examples are provided in Extended Data Fig. 1c). Based on stringent criteria (Methods), we focused on 18 placenta-specific candidate regulatory repeat families, all of which were ERV-associated LTRs (Supplementary Table 1). This included families with known examples of elements bearing promoter activity in the placenta: LTR10A (*NOS3* gene), LTR2B (*PTN* gene), MER39 (*PRL* gene), MER39B (*ENTPD1* gene) and MER21A (*HSD17B1* and *CYP19A1* genes)[19].

We then characterized in more detail the H3K27ac-marked elements from each of these families by performing CUT&Tag for H3K4me1, H3K4me3, H3K9me3 and H3K27me3 in hTSCs. This revealed that a large proportion of H3K27ac-marked elements were also marked by H3K4me1 (median 72%, range 44–85%), a signature of active enhancers (Fig. 1e). Only a small proportion (median 3%, range 0–20%) was marked by H3K4me3, a signature of active promoters (Fig. 1e). There was also a large fraction of elements that were marked by H3K4me1 alone (median 71% of all H3K4me1 elements, range 27–81%), which is normally associated with poised enhancers (Fig. 1e), raising the possibility that this group of elements becomes active following the differentiation of hTSCs. To test this, we differentiated hTSCs into extravillous trophoblast (EVT; Extended Data Fig. 1d) and performed CUT&Tag for H3K27ac. Half of the hTSC-active ERV families remained H3K27ac-enriched in EVT, whereas others were specific to the stem cell state (for example, LTR10A and MER61E; Extended Data Fig. 1e). In line with our hypothesis, a large proportion of ERVs that are active in EVT were in a poised enhancer state in hTSCs (Extended Data Fig. 1f,g). It is possible that other poised ERV enhancers become active upon differentiation into SynT (Extended Data Fig. 1d), but the multinucleated nature of these cells seemingly interfered with our CUT&Tag attempts.

Our analyses suggest that a large number of ERVs (nearly all of which are primate-specific) may act as gene regulatory elements in human trophoblast, showing dynamic changes during differentiation.

### hTSC-active ERVs bind key placental transcription factors

Enhancers function to regulate gene expression through the binding of transcription factors. We therefore identified transcription-factor binding motifs that were enriched within each hTSC-active (that is, marked by H3K27ac) ERV family, and focused on a selection of transcription factors that are expressed in trophoblast (Methods). Reassuringly, we identified previously described motifs on MER41B for signal transducer and activator of transcription (STAT) proteins and serum response factor (SRF; Extended Data Fig. 2a)[12,21]. We also uncovered a large collection of motifs for transcription factors with known roles in trophoblast development. Multiple families bore motifs for key factors involved in the maintenance of the stem cell state, such as ELF5, GATA3, TFAP2C, TP63 and TEAD4 (Fig. 2a and Extended Data Fig. 2a). Additional transcription factors with known roles in placental development and/ or physiology included JUN/FOS[27,28], PPARG/RXRA[29] and FOXO3[30]. Most of these motifs were also present in elements negative for H3K27ac (Extended Data Fig. 2b), suggesting that there are additional genetic or epigenetic determinants of their activity. There were nonetheless exceptions where motifs were only found in active elements, such as GATA3 in LTR2B elements and SRF in MER61D elements (Extended Data Fig. 2b).

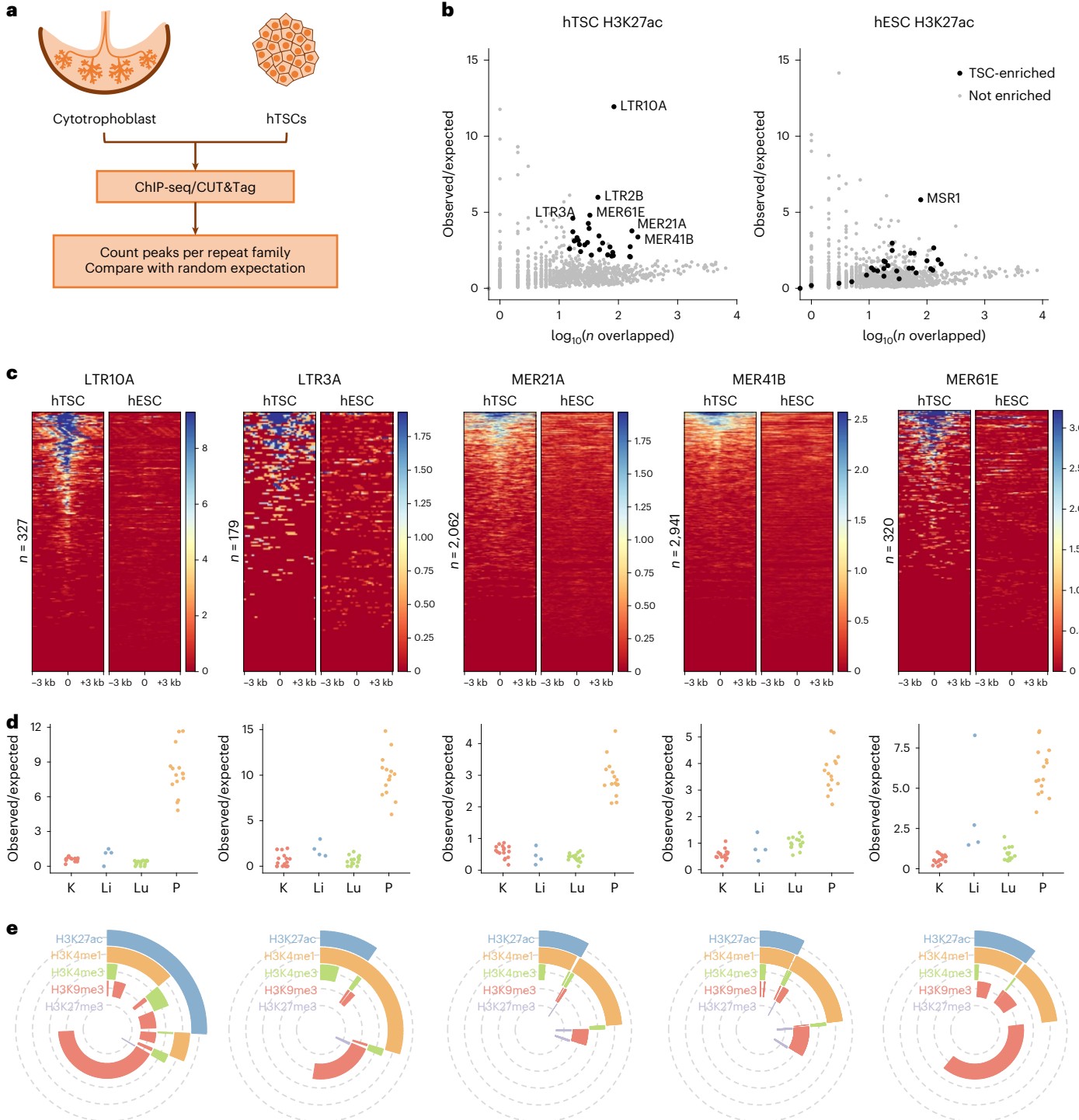

**Fig. 1 | Gene regulatory signatures of ERVs in human trophoblast. a**, Primary cytotrophoblast and hTSCs were profiled for H3K27ac to identify repeat families with putative gene regulatory potential. **b**, Enrichment for H3K27ac peaks for each repeat family in hTSCs or hESCs. Families with significant enrichment in hTSCs are highlighted. **c**, H3K27ac profiles of a subset of hTSC-enriched ERV families in hTSCs and hESCs. Each line represents an element in that family.

**d**, Enrichment for DNase hypersensitive sites in the same ERV families, in the kidney (K), liver (Li), lung (Lu) and placenta (P). Each datapoint represents a different ENCODE dataset. **e**, Proportion of elements from the same ERV families overlapping particular combinations of histone modifications. The data availability statement provides details of source data for **b** and **c**.

To validate the binding of some of these transcription factors to hTSC-active ERVs, we performed CUT&Tag or CUT&RUN[31] for JUN, JUND, GATA3, TEAD4 and TFAP2C (Fig. 2b). We applied the same peak enrichment pipeline as described above (Fig. 1a) and found that several of these families were enriched for one or more of the evaluated transcription factors (Fig. 2c and Extended Data Fig. 2c). In contrast,

TE families active specifically in hESCs showed little to no enrichment of these factors (Fig. 2c and Extended Data Fig. 2c). LTR10A and LTR10F elements were strongly enriched for JUN binding, as predicted from our motif analysis. Similarly, binding to motif-bearing ERVs was confirmed for GATA3 (for example, LTR2B, MER11D and MER61E), TEAD4 (for example, LTR3A, LTR7C and MER41C) and TFAP2C (for example, LTR23

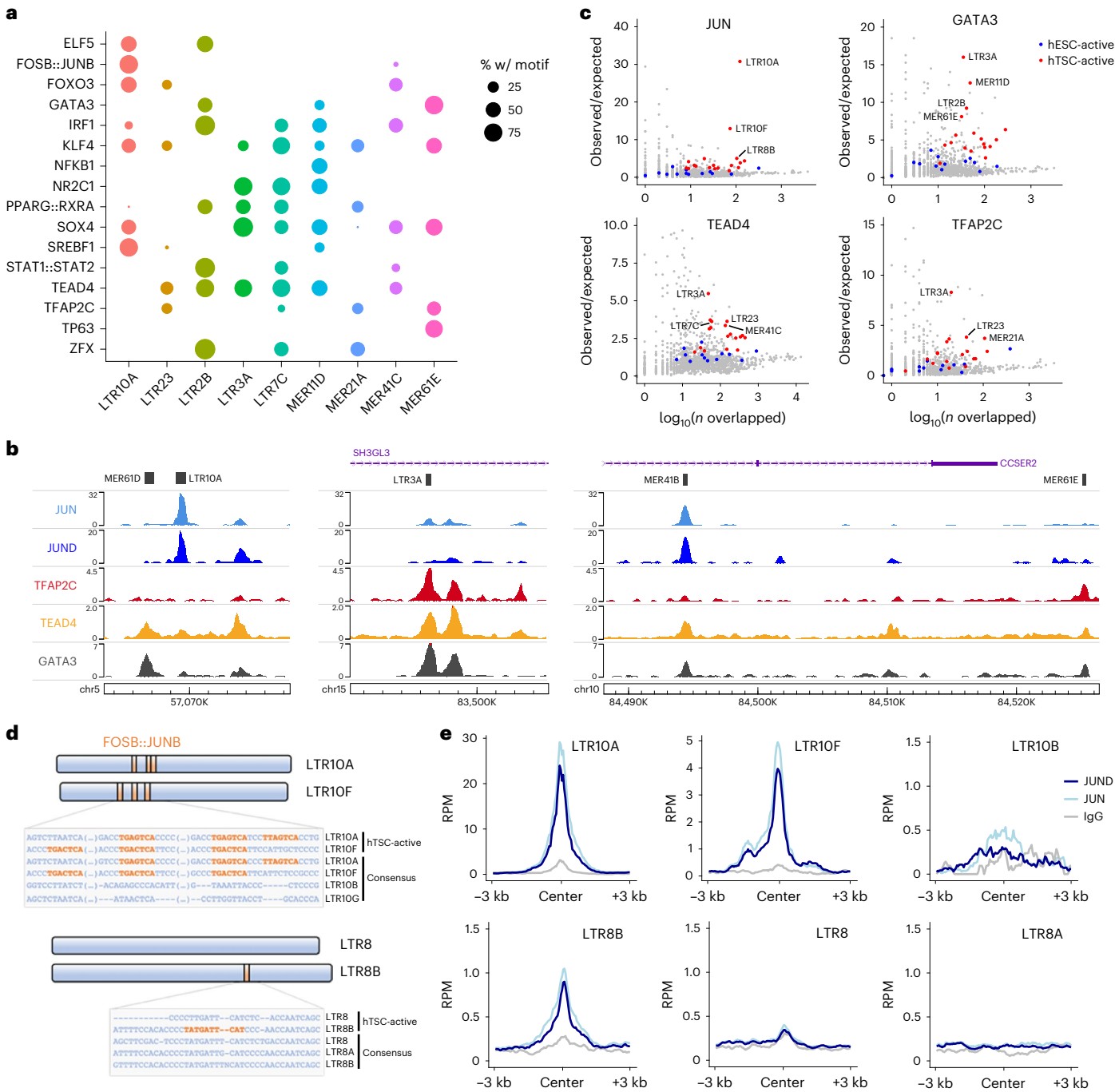

**Fig. 2 | Transcription-factor repertoire at hTSC-active ERVs. a**, Proportion of H3K27ac-marked elements from selected ERV families bearing motifs for the transcription factors on the *y* axis. **b**, Genome browser snapshots of transcription-factor CUT&Tag/RUN data, showing examples of enrichment over ERVs. **c**, Repeat family-wide enrichment for peaks from transcription-factor CUT&Tag/RUN data on hTSCs. H3K27ac-enriched families in hTSCs or hESCs are highlighted. **d**, Schematic and alignment of LTR10 and LTR8 subfamilies showing the presence of FOSB::JUNB (that is, AP-1) motifs in the genome-wide consensus sequence and/or in a consensus of the H3K27ac-marked elements. **e**, Mean CUT&Tag profiles for JUN and JUND over LTR10 and LTR8 subfamilies. See the data availability statement for details on source data for **a**, **c** and **e**.

and MER21A). We also found instances of enriched transcription-factor binding to families that seemingly do not bear the corresponding motif (for example, GATA3 and TFAP2C binding at LTR3A elements). This could reflect limitations of motif-finding approaches and/or suggest indirect transcription-factor recruitment—for example, TEAD4 may recruit TFAP2C[32].

Given the striking enrichment for JUN and JUND binding over LTR10A/F elements, we further explored the corresponding motifs in

these families, as well as in LTR8B. JUN and JUND are two subunits of the activator protein-1 (AP-1) complex, which can heterodimerize with the FOS family of transcription factors. AP-1 plays important roles in cell proliferation and survival, and has been implicated in the regulation of trophoblast differentiation and invasion[27,28]. Both LTR10A and LTR10F active elements contained three AP-1 motifs that were also present in the family-wide consensus sequence, whereas other LTR10-related families lacked any such motifs (Fig. 2d). In strict correspondence

with this, binding was observed for LTR10A/F, but not other LTR10 families (Fig. 2e). JUN/JUND enrichment at LTR10A elements may be substantially more pronounced than for LTR10F due to differences in motif arrangement, cooperative binding with other transcription factors, or chromatin environment. In the case of LTR8B, only active elements contained one AP-1 motif, suggesting divergence of a sub-set of elements after retroviral endogenization (Fig. 2d). CUT&Tag profiles confirmed that JUN/JUND bound LTR8B only, and not other LTR8-related families (Fig. 2e).

These results show that ERV families bearing regulatory potential in human trophoblast are bound by multiple transcription factors that play important gene regulatory roles in trophoblast.

## Genes near active ERVs display biased trophoblast expression

To assess the potential of hTSC-active ERVs to drive trophoblast gene expression, we first asked whether some functioned as gene promoters. We performed de novo transcriptome assembly on our primary cytotrophoblast data[23] and extracted transcripts for which an ERV from the selected families overlapped the transcriptional start site. In line with the relatively small proportion of H3K4me3-containing elements (Fig. 1e), we identified few ERVs with apparent promoter activity (Supplementary Table 2), many of which had been previously reported (promoters for *CYP19A1*[33], *PTN*[34], *PRL*[35] and *MID1*[36]). Another notable gene was *ACKR2*, which encodes a chemokine scavenger whose expression in trophoblast is driven by a MER39 element (Extended Data Fig. 3a), and deficiency of which leads to placental defects and pre/neonatal mortality in mice[37]. Most other transcripts associated with ERV promoters were lowly expressed in cytotrophoblast (Supplementary Table 2).

We next sought to evaluate the regulatory potential of candidate ERV-derived enhancers. Using published RNA-seq data[24], we asked whether the distance to active ERVs correlated with gene expression in trophoblast cells, by comparing it to the expression in placental stroma (non-trophoblast tissue). We found a strong association between gene–ERV distance and preferential expression in both undifferentiated and differentiated trophoblast (Fig. 3a and Extended Data Fig. 3b). In contrast, genes proximal to H3K27ac-marked ERVs in hESCs displayed no preferential expression in trophoblast (Fig. 3a and Extended Data Fig. 3b). We also analyzed RNA-seq data from transdifferentiation experiments of hESCs into hTSC-like cells[38], which showed that genes near hTSC-active ERVs displayed higher expression upon trans-differentiation than those close to hESC-active ERVs (Fig. 3b).

Most genes proximal to hTSC-active ERVs were expressed in all three trophoblast cell types analyzed, with smaller subsets displaying expression in a single trophoblast cell type (Fig. 3c). A relatively large fraction showed increased expression upon differentiation into SynT, including a cluster harboring several genes encoding for pregnancy-specific glycoproteins (PSGs), which are highly expressed in SynT. PSGs are the most abundant conceptus-derived proteins circulating in the maternal blood[39]. Their function in pregnancy remains elusive, although they have been associated with immune responses to pregnancy, and low levels of circulating PSGs are linked to recurrent pregnancy loss, fetal growth restriction and preeclampsia[40–42]. Within the PSG cluster, virtually every H3K27ac peak overlaps either a MER11D or LTR8B element (Fig. 3d), which presumably were already present in the ancestral *PSG* gene before its duplication. These two ERV families are associated with SynT-biased gene expression, mostly due to the *PSG* cluster (Extended Data Fig. 3c). This is in contrast with other families that are associated with genes with biased expression across all three trophoblast cell types analyzed (Extended Data Fig. 3c).

We then leveraged information from our motif analysis to interfere with ERV regulatory activity. Given the enrichment in JUN/JUND binding at LTR10A, LTR10F and LTR8B elements, we treated hTSCs with SP600125, an inhibitor of c-Jun N-terminal kinases (JNKs). Unexpectedly, this led to increased levels of phosphorylated c-Jun in the

nucleus, and the same result was obtained with a second AP-1 inhibitor (Extended Data Fig. 4a). JNK signaling is known to have cell-type specific effects[43], partly due to the opposing roles of JNK1 and JNK2[44]. JNK2 deficiency increases c-Jun phosphorylation and stability[44], potentially explaining our results, as JNK2 is highly expressed in hTSCs (Extended Data Fig. 4b). Irrespective of the mechanism, higher levels of phosphorylated c-Jun are predicted to increase expression of AP-1 target genes. Indeed, RNA-seq of SP600125-treated hTSCs revealed that genes proximal to JUN binding sites were on average upregulated (Fig. 3e and Extended Data Fig. 4d). A large number of upregulated genes are implicated in cell migration (Extended Data Fig. 4c–e), which is in line with observations that SP600125 increases trophoblast cell migration[27]. Finally, we found that LTR10A target genes were upregulated upon SP600125 treatment (Fig. 3f), including *NOS3* (also observed with a second AP-1 inhibitor; Extended Data Fig. 4f), whose placenta-specific gene expression is driven by an LTR10A-derived promoter[45]. Although the majority of LTR10F and LTR8B target genes were also upregulated (Fig. 3f), their low number precluded robust statistical analysis. More prominent effects on LTR10A target genes were expected based on the stronger binding of JUN/JUND to this family (Fig. 2e). Nonetheless, expression of all *PSG* genes was increased by at least twofold, suggesting that JUN regulates this cluster via LTR8B elements. We also found that SP600125 led to upregulation of proviral LTR10A elements, but less so for LTR10F (we found no proviral LTR8B elements), and not for other H3K27ac-enriched families (Extended Data Fig. 4g), providing more direct evidence that AP-1 supports the regulatory activity of LTR10A elements.

## ERVs are associated with species-specific gene expression

Primate evolution has involved dramatic divergence in placental phenotypes, including differences in the cellular arrangement of the feto-maternal interface and the extent of trophoblast invasion into the maternal decidua (Fig. 4a)[46]. Great apes display a unique and deep form of trophoblast invasion[47]. The integration of ERVs with regulatory capacity in trophoblast may have helped to fuel such fast placental evolution across primates. To test this, we first identified ERVs in non-human primates that are orthologous to human elements from each selected family. This showed a wide spread of inter-species differences that are in accordance with the evolutionary age of the selected families (Fig. 4b). We then used published RNA-seq data from rhesus macaque TSCs (macTSCs), which were recently derived using the same culture conditions as for hTSCs[48]. We found that one-to-one orthologous genes close to human-specific ERVs displayed, on average, higher expression in hTSCs than in macTSCs, when compared to genes close to conserved ERVs (Fig. 4c). The majority of non-orthologous elements were from the LTR2B family (as expected from its near absence in macaque) and included the previously characterized placenta-specific promoter of *PTN*[34]. Additional LTR2B-associated genes with human-specific expression included *KCNE3*, which encodes an estrogen-receptor-regulated potassium channel[49], and *STOML2*, which regulates trophoblast proliferation and invasion[50].

We and others have previously shown that *Mus*-specific ERVs also act as distal enhancers in mouse TSCs (mTSCs)[17,18]. We therefore extended our comparative expression analysis by asking whether human- and/or mouse-specific ERVs were associated with increased gene expression in the respective species. Indeed, genes with active ERVs nearby in human but not in mouse displayed higher expression in hTSCs, whereas those close to active mouse ERVs had higher expression in mTSCs (Fig. 4d). Genes with mouse-specific expression included a component of the fibroblast growth factor signaling pathway (*Fgfbp1*), which maintains the stem cell state in mouse but not in human trophoblast, and other mTSC markers (*Duox, Duox2* and *NrOb1*)[51]. Conversely, human-specific ERVs were associated with expression of a Wnt signaling receptor (*FZD5*), a pathway that is important for hTSC derivation[24], and *MMP14*, which is important for trophoblast invasion[52]. We also

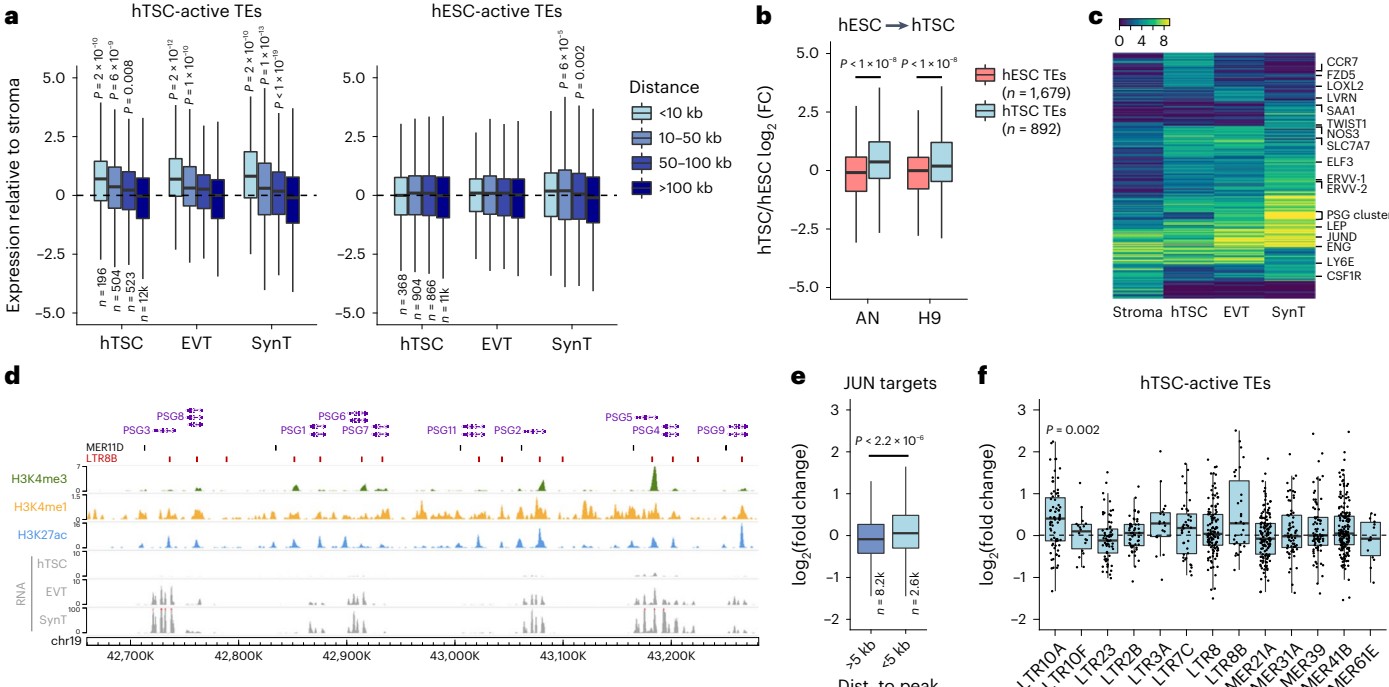

**Fig. 3 | Expression of genes close to regulatory ERVs. a**, Gene expression in hTSCs and hTSC-derived EVT and SynT relative to primary placental stroma. Genes are grouped based on their distance to the nearest H3K27ac-marked TE in hTSCs. The number of genes (*n*) in each set is shown. *P* values are for the difference in comparison to the '>100-kb' group based on the ANOVA and Tukey post-hoc test. **b**, Gene expression in hESC-transdifferentiated hTSCs relative to their respective parental hESC lines (AN or H9). Expression ratios are shown for genes within 50 kb of either hESC- or hTSC-active (H3K27ac-marked) TEs. *P* values are from two-sided Wilcoxon tests with multiple comparisons correction. FC, fold change. **c**, Expression patterns of variably expressed genes across stroma and trophoblast cells. **d**, Genome browser snapshot of the human *PSG*

cluster, highlighting the overlap between H3K27ac peaks and MER11D or LTR8B elements. **e**, FC in gene expression after treatment of hTSCs with JNK inhibitor SP600125, with genes divided according to their distance to the nearest JUN CUT&Tag peak. The number of genes in each set is shown; the *P* value is from a two-sided Wilcoxon test. **f**, FC in expression after SP600125 treatment for genes within 100 kb of an H3K27ac-marked ERV from selected families. The *P* value is from a two-sided Wilcoxon test comparing each distribution to 0, with multiple comparisons correction. The boxplots show median center, 25th and 75th percentile box bounds, and 1.5× interquartile range whisker limits. See the data availability for details of source data for **a–c**, **e** and **f**.

considered potential cases of convergent evolution, whereby the same gene may be regulated by different ERVs in human and mouse. Of the 12 genes that were close to active ERVs in both mouse and human, ten were expressed in mTSCs and hTSCs, including *Zfp42/ZFP42* (Fig. 4e and Supplementary Table 3). Despite being a well-known marker of ESCs, *Zfp42* is also expressed in mouse trophoblast, especially in early embryos[53,54], where it regulates the expression of some imprinted genes[53]. This raises the possibility that different ERVs have convergently been co-opted to maintain the expression of *Zfp42/ZFP42* in mouse and human trophoblast.

These analyses suggest that some of the putative regulatory ERVs that we identified help to drive species-specific expression of genes that are important for trophoblast development and function.

**ERVs regulate genes involved in human trophoblast function**
Previous observations have shown that epigenetic markers are not predictive of gene regulatory activity[18]. To test whether ERVs can act as enhancers in vivo, we utilized CRISPR to genetically excise a subset of candidate regions, and then measured nearby gene expression. Because the efficiency of growing clonal hTSCs from single cells was extremely low, we employed a population-wide lentiviral approach that we had previously used[55], achieving an average of 49% deletion efficiency across different targets and experiments (Supplementary Table 4).

We first excised an enhancer-like MER41B element that is located in the first intron of the *ADAM9* gene (Fig. 5a), which encodes for a metalloproteinase. Genetic variants of *ADAM9* are implicated in preeclampsia[56], and its known substrates play roles in inflammation, angiogenesis,

cellular migration and proliferation[57]. Two independent MER41B excisions were derived, resulting in a 1.7–2-fold decrease in *ADAM9* expression in hTSCs compared to no-sgRNA controls (Fig. 5a). The MER41B LTR is also 8 kb upstream of *TM2D2* and 16 kb upstream of *HTRA4*, a placenta-specific serine peptidase that is upregulated in early-onset preeclampsia[58,59]. Expression of these genes was low and remained largely unchanged following MER41B excision (Extended Data Fig. 5a).

The *CSF1R* gene has a placenta-specific promoter[60], downstream of which lies an enhancer-like LTR10A element (Fig. 5b). Both *CSF1* and *CSF1R* expression increase in the placenta during pregnancy[61]. CSF1 signaling via CSF1R promotes the growth, proliferation and migration of trophoblasts in humans and mice[62–64], and high CSF1 levels are correlated with preeclampsia development[65]. Expression of *CSF1R* was low in undifferentiated hTSCs, but increased following differentiation to EVT, and was highest in SynT-differentiated hTSCs, particularly for the placenta-specific *CSF1R* variant (Fig. 5b). *CSF1R* expression also increased upon SP600125 treatment, suggesting it is regulated by AP-1 (Extended Data Fig. 4f). An excision of the LTR10A was derived in hTSCs, with *CSF1R* expression of the placenta-specific variant being reduced by around twofold in both EVT and SynT cell pools (and to a lesser extend in hTSCs), compared to no-sgRNA controls (Fig. 5b). These differences were not caused by an impairment in trophoblast differentiation efficiency, as judged by the expression of key marker genes (Extended Data Fig. 5c).

We excised a second MER41B element with enhancer-like chromatin conformation (Fig. 5c). The *TWIST1*, *FERD3L* and *HDAC9* genes lie in the vicinity of this LTR, but only *TWIST1* is expressed in hTSCs. TWIST1

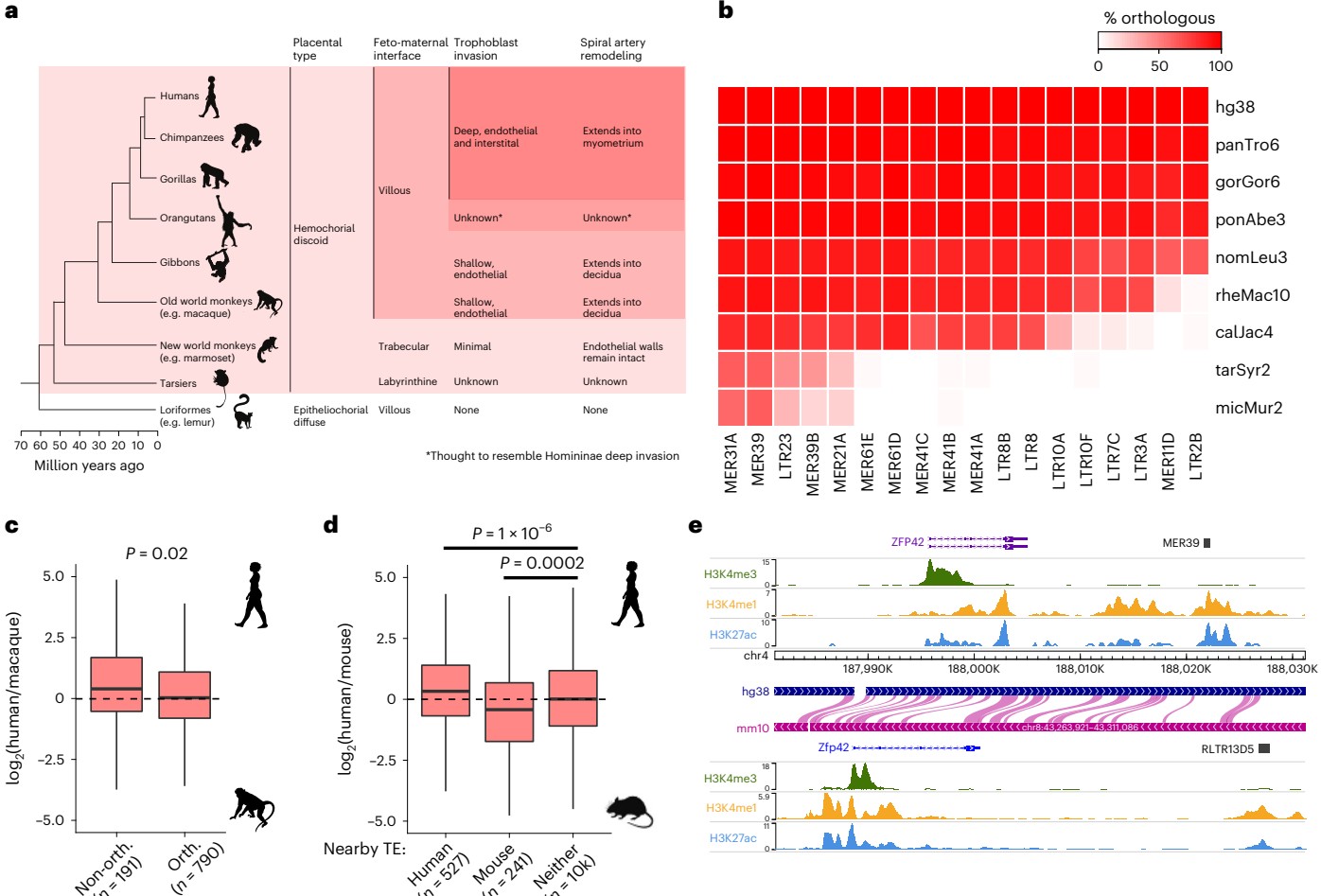

**Fig. 4 | Association of ERVs to species-specific gene expression. a**, Primate phylogeny highlighting cross-species differences in placental morphology and invasion[4,47]. **b**, Proportion of human ERVs from each of the selected families that contain orthologous elements in the non-human primates highlighted in **a**. **c**, Expression difference between human and macTSCs for genes within 100 kb of an H3K27ac-marked ERV from hTSC-active families, depending on whether there is an orthologous element in macaque. The *P* value is from a two-sided Wilcoxon test. **d**, Expression difference between human and mouse TSCs for genes within 50 kb of an H3K27ac-marked ERV in human, mouse (RLTR13D5 and RLTR13B families) or neither species. *P* values are from an ANOVA with Tukey post-hoc test. **e**, Genome browser snapshot showing a putative example of convergent evolution between mouse and human, wherein different enhancer-like ERVs lie downstream of the *ZFP42/Zfp42* gene. Boxplots show median center, 25th and 75th percentile box bounds, and 1.5× interquartile range whisker limits. See the data availability statement for details of the source data for **b**–**d**.

regulates the syncytialization of trophoblast[66,67] and promotes epithelial to mesenchymal transition—a key process in EVT differentiation[68]. We measured the expression of *TWIST1* in MER41B excision hTSC pools and found its expression to be unchanged (Fig. 5c). Because TWIST1 is important for trophoblast differentiation, we also differentiated hTSCs to EVT and SynT, resulting in a 25–41-fold increase in *TWIST1* expression, but there was no difference in *TWIST1* expression in excision versus no-sgRNA differentiated cells (Extended Data Fig. 5b). This particular MER41B element is therefore either a redundant enhancer or does not regulate *TWIST1*, highlighting the importance of these genetic experiments.

As previously mentioned, the *PSG* cluster on chromosome 19 includes MER11D and LTR8B elements at each tandemly repeated gene locus, all featuring enhancer-like chromatin features in hTSCs (Fig. 3d). We excised an LTR8B element within the second intron of one of the most highly expressed PSG in humans, *PSG5* (Fig. 5d)[69], deriving two independent excisions. *PSG5* expression was low in undifferentiated hTSCs and remained unchanged in LTR8B excision pools compared to no-sgRNA controls (Fig. 5d). However, *PSG5* expression was increased following differentiation to EVT and SynT and was reduced in LTR8B excision pools compared to no-sgRNA controls in both trophoblast

types (Fig. 5d), whereas differentiation efficiency was unaffected by the excision (Extended Data Fig. 5d). The fact that the enhancer activities of this LTR8B-*PSG5* element and the LTR10A-*CSF1R* element are most strongly expressed after differentiation supports the notion that some hTSC-active ERVs also play roles in differentiated trophoblast, as suggested by our epigenomic and transcriptomic analyses above.

### An LTR10A-derived enhancer promotes *ENG* expression

We were particularly interested in an enhancer-like LTR10A element within the first intron of the endoglin gene (*ENG/CD105*; Fig. 6a). ENG is a transforming growth factor-beta (TGF-β) 1 and 3 co-receptor, with both membrane-bound and soluble cleavage variants, highly expressed in the endothelium and SynT, and involved in the pathogenesis of preeclampsia[70]. The serum levels of soluble ENG (sENG) are strongly correlated with the severity of preeclampsia[71]. Membrane-bound ENG is also expressed in villous cell columns in the first trimester of pregnancy, regulating trophoblast differentiation to EVT[72,73].

We derived three independent excisions of the *ENG* LTR10A in hTSCs, which resulted in a striking decrease in *ENG* expression compared to no-sgRNA controls (Fig. 6b). Expression of neighboring genes (*AK1* and *FPGS*) did not change (Extended Data Fig. 6a). To confirm

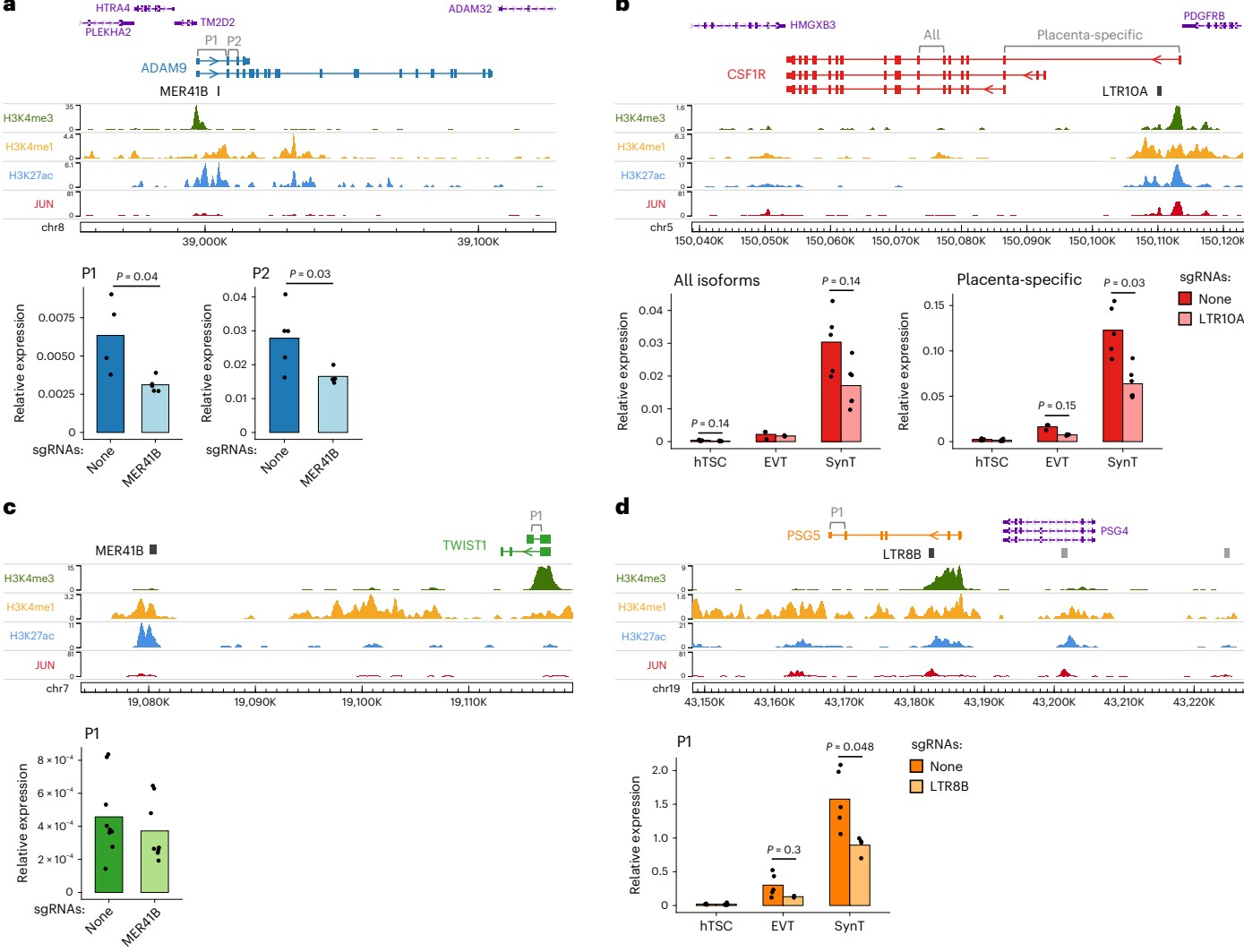

**Fig. 5 | Genetic excision of hTSC-active ERVs. a**, Genome browser snapshot of the *ADAM9* locus containing an enhancer-like MER41B element, and *ADAM9* expression (using primer pairs highlighted on the gene annotation) in hTSC populations treated with lentiviral CRISPR constructs carrying either no sgRNAs (*n* = 4) or sgRNAs that excise the MER41B element (*n* = 5 from two sgRNA sets). **b**, As in **a**, but for *CSF1R* and its intronic LTR10A elements. *CSF1R* expression was measured in hTSCs (*n* = 5 from one sgRNA set) and hTSC-derived EVT (*n* = 3) and SynT (*n* = 5 no sgRNA, *n* = 6 LTR10A sgRNAs). **c**, As in **a**, but for *TWIST1* and a downstream MER41B element (*n* = 9 no sgRNA, *n* = 8 MER41B sgRNAs; from three infections using one sgRNA set). **d**, As in **a**, but for *PSG5* and its intronic LTR8B elements. *PSG5* expression was measured in hTSCs (*n* = 3 no sgRNA, *n* = 6 LTR8B sgRNAs from two sgRNA sets) and hTSC-derived EVT (*n* = 5 no sgRNA, *n* = 2 LTR8B sgRNAs) and SynT (*n* = 5 no sgRNA, *n* = 4 LTR8B sgRNAs). *P* values throughout are from two-sided Wilcoxon tests with multiple comparisons correction. See the data availability statement for details of the source data for RT–qPCR.

that loss of *ENG* expression was strictly associated with deletion of the LTR10A element, and not transcriptional interference by the CRISPR–Cas9 machinery, we also deleted a control region upstream of the putative LTR10A enhancer that was devoid of active histone modification marks in hTSCs (Fig. 6a). Reassuringly, deletion of this region had no effect on *ENG* expression (Fig. 6b). In line with the role of AP-1 in regulating LTR10A regulatory activity, increased c-Jun phosphorylation led to *ENG* upregulation (Extended Data Fig. 4f). To assess the phenotypic impact of LTR10A deletion, we first measured cell proliferation in hTSCs, finding no difference when compared to no-sgRNA controls (Fig. 6c). Differentiation to EVT was not affected by LTR10A deletion, as measured by the percentage of human leukocyte antigen G (HLAG)-positive cells (Fig. 6d and Extended Data Fig. 6b). LTR10A-deleted EVT were also morphologically similar to control EVT and retained the capacity to invade Matrigel in a transwell assay (Extended Data Fig. 6c), although technical variation precluded a more quantitative assessment of the extent of invasion. Notably, ENG

expression remained lower (albeit variable) in LTR10A-deleted EVT, when compared with no-sgRNA controls (Fig. 6d). Differentiation into SynT was also unaffected by deletion of the LTR10A element (Extended Data Fig. 6d). Despite a large rise in *ENG* levels following SynT differentiation, LTR10A-deleted cells expressed less *ENG* than no-sgRNA controls (Fig. 6e). We therefore tested whether sENG protein levels were impacted by the LTR10A enhancer by performing enzyme-linked immunosorbent assays (ELISA) in the media of SynT cultures. We detected sENG at concentrations varying from ~100 pg ml$^{-1}$ to 600 pg ml$^{-1}$ and observed a significant decrease in sENG protein levels in LTR10A-*ENG* excised SynT cultures when compared with no-sgRNA control (Fig. 6e).

These experiments raise the possibility that deregulation of the *ENG*-LTR10A element may be associated with elevated levels of *ENG* in preeclampsia. We leveraged recently published H3K27ac ChIP-seq data from cytotrophoblast isolated from preeclamptic or uncomplicated pregnancies[74]. In uncomplicated pregnancies, there was clear H3K27ac enrichment over the LTR10A-*ENG* element in second-trimester

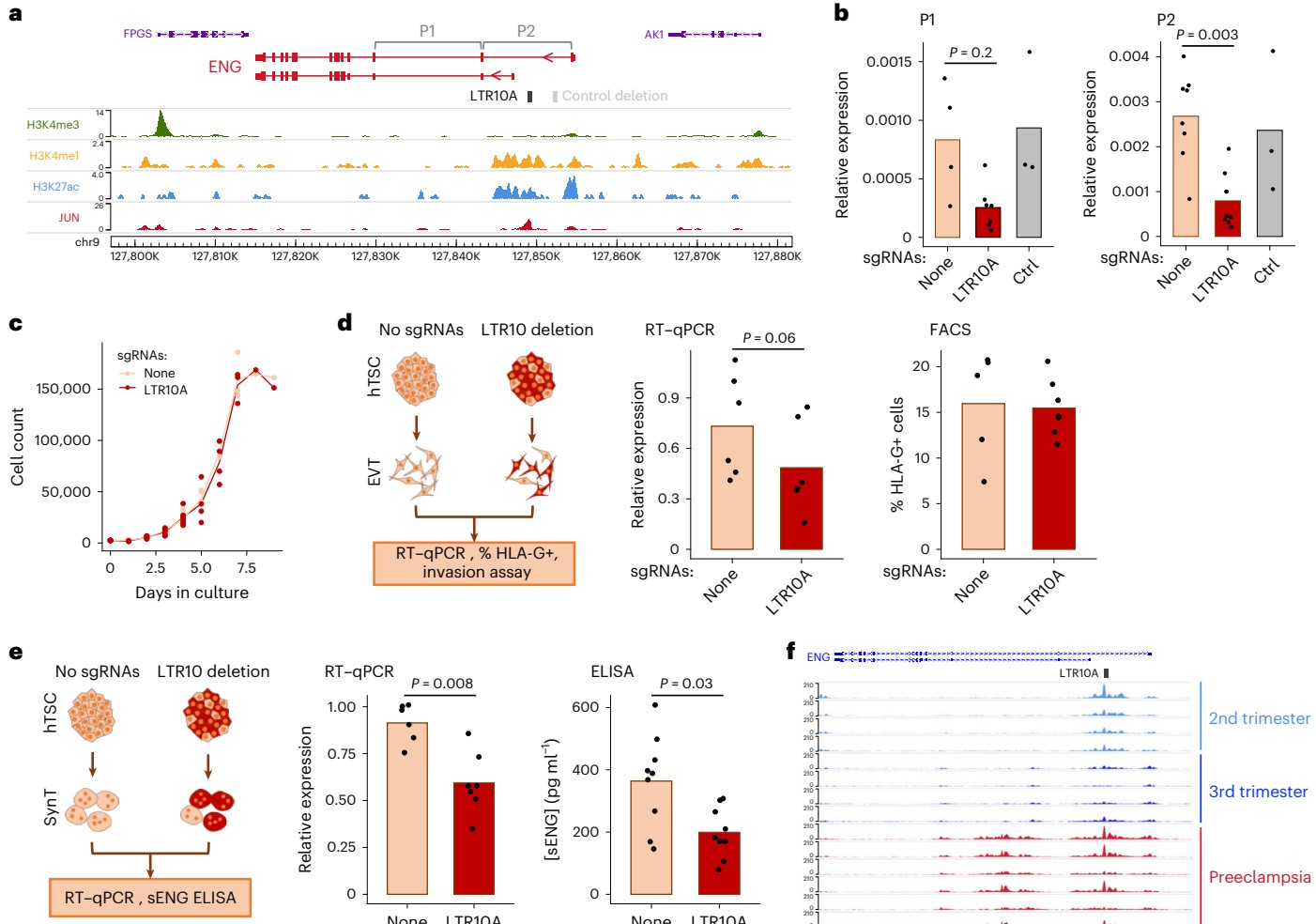

**Fig. 6 | Impact of an LTR10A element within the *ENG* gene. a**, Genome browser snapshot of the *ENG* locus containing an enhancer-like LTR10A element in its first intron. **b**, *ENG* expression (using the primer pairs highlighted in **a**) in hTSC populations treated with lentiviral CRISPR constructs carrying no sgRNAs (*n* = 4 P1, *n* = 8 P2), sgRNAs that excise the LTR10A element (*n* = 7 P1, *n* = 9 P2 from three sgRNA sets) or sgRNAs that excise a control region highlighted in **a** (*n* = 3). **c**, Growth curves for hTSC populations carrying no sgRNAs or LTR10A-excising sgRNAS. **d**, The same hTSC populations were differentiated to EVT, and assays were performed for *ENG* RNA levels (*n* = 6) and HLAG cell surface expression

(*n* = 5 no sgRNAs, *n* = 7 LTR10A sgRNAs). **e**, The same hTSC populations were differentiated to SynT, and assays were performed for *ENG* RNA levels (*n* = 6 no sgRNAs, *n* = 7 LTR10A sgRNAs) and secretion of soluble ENG protein (*n* = 9). **f**, Genome browser snapshot of the *ENG* locus with cytotrophoblast H3K27ac ChIP-seq data from second- or third-trimester uncomplicated pregnancies, or third-trimester severe preeclampsia placentas. *P* values throughout are from two-sided Wilcoxon tests with multiple comparisons correction where relevant. See the data availability statement for details of the source data for **b**–**e**.

placentas, which decreased in third-trimester placentas (Fig. 6f). However, in third-trimester placentas from severe preeclampsia, H3K27ac levels remained high over the LTR10A-*ENG* enhancer (Fig. 6f), suggesting that this ERV may help to maintain ENG expression unduly high in preeclampsia.

These data show that the LTR10A-*ENG* element acts as an enhancer in human trophoblast, regulating *ENG* expression and sENG protein production, with potential implications for preeclampsia.

## Discussion

We have identified multiple ERV families that are enriched for a chromatin signature of *cis* regulatory elements in human trophoblast, most resembling enhancers. Our stringent criteria ensured that these regulatory profiles reflect what is observed in vivo and are not driven by non-trophoblast cell types present in the placenta. Indeed, we find that the activity of these families is largely trophoblast-specific, with several playing important roles in both undifferentiated and differentiated trophoblast cell types. Our genetic editing experiments have demonstrated that at least a subset of the ERVs we identified act as bona fide

enhancers of genes, with important roles in placentation. Notably, the fact that we were limited to performing CRISPR on a population scale implies that the effects we observed are actually an underestimation of the true importance of those ERVs to gene expression.

When we compared our selected families with those identified by the Macfarlan laboratory as being enriched for lineage-specific placental enhancers, we found multiple ERV families in common, including MER21A, MER41B, LTR8 and MER39[21]. On the other hand, the study by Sun et al.[21] study did not list other ERV families identified here, most prominently LTR10A, which has a strong regulatory signature and copies of which we demonstrated act as enhancers of important placental genes. Conversely, we find no evidence of regulatory activity from the MaLR group of ERV families identified by Sun and colleagues. One possible reason for these discrepancies is that Sun et al. used whole placental explants, leading to a mixed epigenomic profile from multiple cell types.

We also noted that a number of our trophoblast-active ERV families were recently identified as active in several cancers, including LTR10A, LTR10F, LTR2B and MER11D[75]. Multiple parallels have previously been

drawn between trophoblast and cancer cells, including an epigenetic landscape that is strikingly different from other differentiated cell types[76]. The combination of an arguably permissive chromatin conformation and shared signaling pathways[77] may make the co-option of ERVs for both placental development/function and cancer a frequent occurrence. For example, the Hippo signaling pathway, acting through YAP and TEAD4, plays key roles in both tumorigenesis and placental development[78]. The activation of LTR10A elements in both contexts is also seemingly driven by a shared signaling pathway, that is, MAPK/AP-1[75].

Several specific elements identified here suggested a potential important role for ERVs in placental evolution, including the multiple LTR8B and MER11D copies associated with the *PSG* gene cluster. The *PSG* cluster is semi-conserved in primates, with 6 to 24 genes found in Old World monkeys, one to seven genes in New World monkeys and none in more distantly related primates such as lemurs, suggesting that the presence of *PSG* may correlate with hemochorial placentae, given that lemurs have an epitheliochorial placenta[79]. Notably, *PSG* clusters are also present in mice, which also have a hemochorial placenta, and this region was expanded independently in mice and primates, suggesting convergent evolution[80]. Our results suggest that the integration of LTR8B elements ahead of *PSG* cluster expansion in humans was an important step that contributed to high trophoblast expression of these genes. MER11D elements may play a similar role and, together with LTR8B, be responsible for much of the transcriptional regulation of this important locus. Similarly, MER61D/E retrotransposition may have played a key role in setting the TP63 binding landscape, which in human trophoblast supports cell proliferation and prevents cellular differentiation in trophoblast[81]. TP63 belongs to the same family of transcription factors as TP53, sharing many of its binding sites[82]. It was previously shown that MER61 elements expanded the TP53 binding network in primates, and a number of copies have been exapted to mediate cellular stress responses in lymphoblastoid cells[83].

Other examples suggest a role of ERVs in coordinating the expression of genes from the same pathway. MER21A elements were previously shown to act as promoters of the steroidogenesis pathway genes *CYP19A1* and *HSD17B1*, implicating these LTRs in the regulation of steroidogenesis in human trophoblast. We also noted that both *NOS3* and *ENG* bear LTR10A elements as major transcriptional regulators. Interestingly, the contribution of sENG to vascular pathology in preeclampsia is partially due to effects on NOS3; in concert with FLT1, ENG reduces placental angiogenesis and vasodilation of maternal spiral arteries, and increases vessel permeability[70,84].

Our results implicate regulatory ERVs in pregnancy complications, prompting the need for further investigation of their functional impact on pregnancy outcomes. These ERVs may bear genetic variants that are difficult to investigate due to their repetitive nature, and that may affect their regulatory activity in the placenta. Notably, structural variants in LTR10A/F elements (in the form of variable number tandem repeats), some potentially contributing to cancer, were recently described[75]. Research into the effects of such variants in the placenta will benefit from more complex models that can assess the impact of regulatory ERVs on cell–cell interactions, such as those between the conceptus and the mother, that make pregnancy so unique. Such experiments will be greatly supported by the recent development of placental and endometrial organoids, as well as platforms that support the study of trophoblast invasion[85,86].

## Online content

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

## Methods

### Tissue culture

hTSCs were cultured according to ref. 24, with modifications as outlined in ref. 87. Briefly, ~1 × 10⁵ hTSCs were seeded onto six-well plates coated with either 5–10 µg ml⁻¹ collagen IV (10376931, Fisher Scientific) or 0.5 µg ml⁻¹ iMatrix 511 (NP892-011, Generon). The basal TS medium comprised DMEM/F12 (Gibco) supplemented with 1% KnockOut serum replacement (KSR; Life Technologies Ltd Invitrogen Division), 0.5% penicillin-streptomycin (Pen-Strep, Gibco), 0.15% BSA (A9205, Sigma-Aldrich), 1% ITS-X supplement (Fisher Scientific) and 200 µM l-ascorbic acid (Sigma-Aldrich). Complete medium contained 2.5 µM Rho-associated protein kinase (ROCK) inhibitor Y27632, TGF-B inhibitors A83-01 (5 µM), CHIR99021 (2 µM) (all StemMACS) and epidermal growth factor (50 ng ml⁻¹, E9644, Sigma-Aldrich), with 0.8 mM histone deacetylase inhibitor valproic acid (PHR1061, Sigma-Aldrich). The hTSCs were passaged using TrypLE Express (Fisher Scientific) and split 1:3 to 1:5 every 2–3 days. All cells were cultured at 37 °C in 5% $CO_2$ and 20% $O_2$. For JNK inhibition experiments, hTSCs were seeded onto 12-well or six-well plates and treated with either 10 µM SP600125 (Cambridge Bioscience) for 2–4 days or 20 µM SR11302 (Cambridge Bioscience) for 1–2 days. Control wells were treated with DMSO. Changes in the expression of diagnostic genes were consistent across different cell-collection time points.

### Differentiation to EVT and SynT

Differentiation was performed as outlined in ref. 24. For EVT, hTSCs were seeded in a six-well plate coated with 1 µg ml⁻¹ Col IV at a density of 0.75 × 10⁵ cells per well and cultured in 2 ml of EVT medium: DMEM/F12 supplemented with 0.1 mM 2-mercaptoethanol, 0.5% Pen-Strep, 0.3% BSA, 1% ITS-X, 100 ng ml⁻¹ human neuregulin-1 (NRG1; Cell Signaling), 7.5 µM A83-01, 2.5 µM ROCK inhibitor Y27632 and 4% KSR. Cells were suspended in the medium, and Matrigel (LDEV-free, 354234, Scientific Laboratory Supplies) was added to a final concentration of 2%. On day 3, the medium was replaced with EVT medium without NRG1, and Matrigel was added to a final concentration of 0.5%. On day 6, the medium was replaced with EVT medium without NRG1 and KSR, and Matrigel was added to a final concentration of 0.5%, with cells grown for a further two days. Differentiation efficiency was measured using fluorescence-activated cell sorting (FACS) following staining with an allophycocyanin (APC) conjugated antibody to HLAG (APC anti-human HLAG antibody clone: 87G, BioLegend UK); in unmodified hTSCs, this was consistently between 40 and 70%, and in lentivirus-infected CRISPR excision lines and controls, between 10 and 30%. SynT (three-dimensional) differentiation was carried out by seeding 2.5 × 10⁵ hTSCs in uncoated six-well plates, cultured in 2 ml of SynT medium: DMEM/F12 supplemented with 0.1 mM 2-mercaptoethanol, 0.5% Pen-Strep, 0.3% BSA, 1% ITS-X, 2.5 µM ROCK inhibitor Y27632, 50 ng ml⁻¹ epidermal growth factor, 2 µM forskolin (Cambridge Bioscience) and 4% KSR. A 2-ml volume of fresh ST(3D) medium was added on day 3. Cells were passed through a 40-µm mesh strainer, and cells remaining on the strainer were collected and analyzed. Differentiation was assessed using quantitative polymerase chain reaction with reverse transcription (RT–qPCR) and immunostaining (the primers used are listed in Supplementary Table 5).

### FACS

Cells were single-cell-dissociated using TrypLE Express and washed in FACS buffer (PBS supplemented with 4% KSR). The cells were then resuspended in 500 µl of fresh FACS buffer, passed through a 70-µm cell strainer, and GFP+ (green fluorescent protein) cells collected and replated (lentiviral infections). To sort EVT-differentiated cells, 400 µl of the cell suspension was incubated with APC-HLAG for 15 min in the fridge, then 100 µl was used as an unstained control. Following antibody incubation, the cells were washed twice with FACS buffer, resuspended in fresh FACS buffer, and passed through a 70-µm cell strainer. Flow cytometry was performed using a FACS Aria II system, and the data were analyzed with FlowJo software.

### Immunostaining

Five thousand hTSCs per well were grown on collagen-IV-coated glass coverslips in a 24-well plate, and the differentiation protocol to EVT followed. On day 8 of EVT differentiation, the medium was removed and the cells washed with PBS three times, before fixing in 4% paraformaldehyde for 10 min then staining. Differentiated SynT cells were centrifuged briefly (300g, 1 min), resuspended gently in 500 µl PBS + 4% KSR, and added dropwise to poly-L-lysine-coated glass coverslips. Once SynT cells had gathered on the coverslip, the PBS + KSR was removed and cells were fixed in 4% paraformaldehyde for 10 min. Both fixed EVT and SynT cells were permeabilized with 0.1% Triton X in blocking buffer (1% BSA (wt/vol) 2% FCS (vol/vol)) for 5 min, followed by 1 h of incubation in blocking buffer. Incubation with the primary antibody (SDC-1 (1:200, CD138 mouse anti-human, phycoerythrin, Clone: MI15, Fisher Scientific, UK) for SynT and HLAG (1:100, APC anti-human HLAG antibody clone: 87G, BioLegend UK) for EVT) diluted in blocking buffer was performed at 4 °C overnight. The secondary goat anti-mouse IgG, Alexa Fluor Plus 488 secondary antibody (cat. no. A32723, Thermo Fisher Scientific) was added at a 1:250 dilution for 1 h at room temperature. 4′,6-Diamidino-2-phenylindole (300 nM) was used as a nuclear stain. Images were collected using a Leica DM4000 epifluorescence microscope.

### Western blot

Cells were washed with ice-cold PBS, lysed with ice-cold 'Triton lysis buffer' (50 mM Tris, pH 7.5, 150 mM NaCl, 1% Triton X-100, supplemented with protease inhibitor cocktail, PMSF and sodium orthovanadate) and pelleted. Supernatants were labeled as 'Triton-soluble fractions', which are primarily cytoplasmic. Pellets were washed with ice-cold PBS and lysed with 2% sodium dodecyl sulfate (SDS), followed by sonication (these were labeled as 'Triton-insoluble fractions', which are primarily nuclear). Protein concentrations were assessed using a Pierce BCA protein assay kit (Thermo Fisher Scientific). Samples were separated by SDS–polyacrylamide gel electrophoresis (PAGE) and transferred into nitrocellulose membranes. Membranes were incubated with 5% skimmed milk in PBS-T for 30 min at room temperature, followed by overnight incubation with the following antibodies at +4 °C: anti pSer63 c-Jun (Cell Signalling, cat. no. 2361, 1:1,000) and anti c-Jun (Cell Signalling, cat. no. 9165, 1:1,000). After washing with PBS-T, membranes were incubated with peroxidase-conjugated anti-rabbit IgG (Sigma-Aldrich, A6154, 1:10,000) for 1 h at room temperature, washed with PBS-T and exposed to ECL reagent (Sigma-Aldrich, cat no. WBKLS0500). Signals were visualized using X-ray film and a film processor (pSer63 c-Jun) or using a ChemiDoc MP Imaging System (Bio-Rad). Membranes were re-incubated with anti α-tubulin (Sigma-Aldrich, T9026, 1:20,000) or anti H3 (Abcam, ab1791, 1:10,000) for 1 h at room temperature, washed, and then incubated for 1 h with peroxidase-conjugated anti-mouse IgG (Sigma-Aldrich, A0168, 1:10,000) or anti-rabbit IgG, respectively, followed by visualization using ChemiDoc.

### ChIP-seq

hTSCs were dissociated using TrypLE Express, pelleted, and washed twice with PBS. The cell pellets were fixed with 1% formaldehyde for 12 min in PBS, followed by quenching with glycine (final concentration of 0.125 M) and washing. Chromatin was sonicated using a Bioruptor Pico device (Diagenode) to an average size of 200–700 bp. Immunoprecipitation was performed using 10 µg of chromatin and 2.5 µg of human antibody (H3K4me3 (RRID:AB_2616052, Diagenode C15410003), H3K4me1 (RRID:AB_306847, Abcam ab8895) and H3K27ac (RRID:AB_2637079, Diagenode C15410196)). Final DNA purification was performed using a GeneJET PCR purification kit (Thermo Scientific, K0701) and eluted in 80 µl of elution buffer. ChIP-seq libraries

were prepared from 1 to 5 ng eluted DNA using an NEBNext Ultra II DNA library Prep Kit (New England Biolabs) with 12 cycles of library amplification.

## CUT&Tag

CUT&Tag was carried out as in ref. 22. A total of 100,000 hTS or 50,000 EvT cells per antibody were collected fresh using TrypLE Express and centrifuged for 3 min at 600g at room temperature. Cells were washed twice in 1.5 ml of wash buffer (20 mM HEPES pH 7.5; 150 mM NaCl; 0.5 mM spermidine; 1× protease inhibitor cocktail; Roche 11836170001 (PIC)) by gentle pipetting. BioMagPlus concanavalin A-coated magnetic beads (Generon) were activated by washing and resuspension in binding buffer (20 mM HEPES pH 7.5, 10 mM KCl, 1 mM CaCl$_2$, 1 mM MnCl$_2$), then 10 µl of activated beads were added per sample and incubated at room temperature for 15 min. A magnet stand was used to isolate bead–cell complexes (henceforth 'cells'), the supernatant was removed, and the cells were resuspended in 50–100 µl of antibody buffer (20 mM HEPES pH 7.5; 150 mM NaCl; 0.5 mM spermidine; 1× PIC; 0.05% digitonin, 2 mM EDTA, 0.1% BSA) and a 1:50 dilution of primary antibody (H3K27Ac (39034), active motif, H3K9me3 (C15410193) and H3K27me3 (C15410195), Diagenode; c-Jun (60A8)–9165T, and JunD (D17G2) – 5000S, Cell Signalling; GATA3 sc-268 and TFAP2C sc-12762, Santa Cruz; TEAD4 CSB-PA618010LA01HU, Stratech; negative control rabbit IgG, sc-2027, Santa Cruz). Primary antibody incubation was performed on a Nutator shaker for 2 h at room temperature or overnight at 4 °C. Primary antibody was removed and secondary antibody (guinea pig anti-rabbit IgG (heavy and light chain) ABIN101961) diluted 1:50 in 50–100 µl of Dig-Wash buffer, and added to the cells for 30 min incubation at room temperature. The supernatant was removed and cells were washed three times in Dig-Wash buffer. Protein A-Tn5 transposase fusion protein (pA-Tn5) loaded with Illumina NEXTERA adapters (a kind gift from the Henikoff laboratory, via the Madapura laboratory, diluted 1:250, or Epicypher CUTANA pAG-Tn5, diluted 1:10) was prepared in Dig-300 buffer (0.05% digitonin, 20 mM HEPES, pH 7.5, 300 mM NaCl, 0.5 mM spermidine, 1× PIC) and 50–100 µl was added to the cells with gentle vortexing, followed by incubation at room temperature for 1 h. Cells were washed three times in 800 µl of Dig-300 buffer to remove unbound pA-Tn5. Next, the cells were resuspended in 50–100 µl tagmentation buffer (10 mM MgCl$_2$ in Dig-300 buffer) and incubated at 37 °C for 1 h. To stop tagmentation, 2.25 µl of 0.5 M EDTA, 2.75 µl of 10% SDS and 0.5 µl of 20 mg ml$^{-1}$ Proteinase K was added to the sample and incubated overnight at 37 °C. To extract the DNA, 300 µl phenol:chloroform:isoamyl alcohol (PCI 25:24:1, vol/vol; Sigma-Aldrich) was added to the sample and vortexed. Samples were added to five PRIME Phase Lock Gel light tubes and centrifuged for 3 min at 16,000g. The samples were washed in chloroform, centrifuged for 3 min at 16,000g, and supernatant was added to 100% ethanol, chilled on ice and centrifuged at 16,000g for 15 min. Pellets were washed in 100% ethanol and allowed to air-dry, followed by resuspension in 30 µl of 10 mM Tris-HCl pH 8 1 mM EDTA containing 1/400 RNAse A. Libraries were indexed and amplified, using 21 µl of DNA per sample and adding 25 µl NEBNext HiFi 2× PCR Master mix + 2 µl Universal i5 primer (10 µM) + 2 µl uniquely barcoded i7 primers (10 µM) in 0.2-ml PCR tube strips, using a different barcode for each sample. Cycling parameters were 72 °C for 5 min, 98 °C for 30 s, then 12 cycles of 98 °C for 10 s, 63 °C for 10 s, followed by 72 °C for 1 min, followed by purification with 1X Agencourt AMPure XP beads as per the manufacturer's instructions (Beckman Coulter).

## Cleavage under targets and release using nuclease

CUT&RUN was carried out as in ref. 31. A total of 500,000 hTSCs were washed and resuspended in wash buffer (20 mM HEPES pH 7.5, 150 mM NaCl, 0.5 mM spermidine, plus PIC). Following concanavalin A bead preparation (as for CUT&Tag), the cells were incubated with 50 µl of bead slurry for 15 min and the supernatant removed. Cell–bead

complexes (hereafter, 'cells') were incubated overnight with antibody buffer (as for CUT&Tag) and primary antibodies (H3K27Ac (39034), Active Motif, H3K9me3 (C15410193) and H3K27me3 (C15410195), Diagenode; c-Jun (60A8) – 9165T and JunD (D17G2) – 5000S, Cell Signalling; GATA3 sc-268 and TFAP2C sc-12762, Santa Cruz; TEAD4 CSB-PA618010LA01HU, Stratech and negative control rabbit IgG, sc-2027, Santa Cruz). Primary antibody was removed and the cells were washed three times in Dig-Wash buffer and incubated with 1:200 pA-MNase (a gift from the Hurd laboratory) for 1 h at 4 °C. The cells were washed three times in Dig-Wash buffer, followed by digestion (adding 2 µl 100 mM CaCl$_2$ to 100 µl of sample) for 30 min at 0 °C, and the reaction stopped in STOP buffer (170 mM NaCl, 20 mM EGTA, 0.05% digitonin, 100 µg ml$^{-1}$ RNAse A, 50 µg ml$^{-1}$ glycogen) for 30 min at 37 °C. DNA was extracted using phenol/chloroform, as for CUT&Tag, and barcoded libraries constructed with the NEBNext Ultra II DNA Library Prep Kit for Illumina, using 12 cycles of PCR, followed by purification with 1X Agencourt AMPure XP beads.

## RNA isolation and RT–qPCR

Total RNA was extracted with the AllPrep DNA/RNA Mini Kit (Qiagen) and treated with a TURBO DNA-free Kit (Ambion, AM1907) to remove contaminating genomic DNA. RNA (200 ng) was retrotranscribed using Revertaid reverse transcriptase (Thermo Scientific, EP0441), and cDNA was diluted 1/50 for qPCRs using KAPA SYBR FAST (Sigma-Aldrich, KK4610). RT–qPCR was carried out on a Roche LC480 for 40–45 cycles.

## RNA-seq

Before library construction, 10–100 ng of total RNA was treated with the NEBNext rRNA Depletion Kit. Library construction was performed with the NEBNext Ultra II Directional RNA Library Prep Kit for Illumina, according to the manufacturer's protocol. RNA concentration and integrity were assessed using a Bioanalyzer 2100 system (Agilent Technologies). All hTSC samples had an RNA integrity number equivalent (RINe) value of >9.

## Guide RNA cloning into CRISPR–Cas9 plasmids

For CRISPR/Cas9 deletion of LTRs, sgRNA oligonucleotides (Integrated DNA Technologies) were designed to target upstream and downstream of the LTRs of interest, using Benchling (https://benchling.com) and annealed. Either the upstream or the downstream guide was cloned into plasmid LRG (Lenti_sgRNA_EFS_GFP; deposited by C. Vakoc (Addgene 65656), which expresses GFP. The other guide was cloned into lentiC-RISPR v2, deposited by F. Zhang (Addgene 52961). Clones were verified by Sanger sequencing (Source Bioscience). Guide sequences are listed in Supplementary Table 6.

## Lentivirus-mediated hTSC transduction and selection

Lentivirus was produced in 293T cells by quadruple transfection with CRISPR/Cas9 delivery vectors (see above) and the packaging plasmids psPAX2, (deposited by D. Trono, Addgene 12260) and pMD2.G (deposited by D. Trono, Addgene 12259) using FuGENE HD transfection reagent (Promega E2311). Viral supernatant was collected at 48 h and again at 72 h post transfection, pooled, filtered through 0.45 µm and either used fresh or aliquoted and stored at −80 °C. hTSCs were transduced with lentiviral supernatant supplemented with 4 µg ml$^{-1}$ polybrene for 6 h. Supernatant from lentivirus transfected with psPAX2, pMDG.2, empty LRG and empty lentiCRISPR v2 (that is, no gRNAs) was used alongside each new hTSC infection as a no-sgRNA control. At 48 h after transduction, GFP+ cells were sorted on a FACS Aria II system and replated onto 6- or 12-well plates depending on the cell number. At 24–48 h after sorting, cells were treated with puromycin sulfate for 48 h. DNA and RNA were extracted from the resultant cell populations and genotyped for the presence of excisions. Successful excision pools were further analyzed for the percent of excised alleles and for the expression of genes nearby the targeted LTRs using qPCR and RT–qPCR. The number

of independent RT–qPCR replicates used is visible on the respective plots—these were derived using different sgRNA sets or independent infections (as detailed in the figure legends and Supplementary Table 4), and/or from RNA collections at different passage numbers. Genotyping and RT–qPCR primers are listed in Supplementary Table 5.

## Endoglin ELISA
hTSC pools containing the LTR10A excision at ENG were seeded in parallel with no-sgRNA controls at $2.5 \times 10^5$ cells per well of a six-well plate and differentiated to SynT, as described above. On day 6, medium was collected and stored at −80 °C. SynT cells were snap-frozen for RT–qPCR analysis of differentiation markers and ENG expression. Media aliquots were frozen and subjected to ENG ELISA analysis with the human endoglin ELISA kit (Sigma-Aldrich, RAB0171) according to the manufacturer's instructions. Absorbance was measured at 450 nm.

## Transwell invasion assay
hTSC pools containing the LTR10A excision at ENG were seeded in parallel with no-sgRNA controls at $2.5 \times 10^5$ cells per well of a collagen-IV-coated six-well plate and differentiated to EVT as described above. On days 7–9 of differentiation, cells were treated with TrypLE and counted. A subset (~5%) of cells were tested for differentiation efficiency by HLAG staining and FACS to ascertain the proportion of HLAG+ cells, as described above. Invasion assays were carried out on either inserts coated with 75 µl of Matrigel Basement Membrane Matrix (Fisher Scientific, cat. no. 11573620) diluted 1:1 with serum-free EVT medium; or commercial Matrigel pre-coated inserts (BioCoat Matrigel Invasion Chamber, VWR, cat. no. 734-1047), rehydrated by the addition of 500 µl of serum-free D6 EVT medium in both upper and lower chambers for >1 h at 37 °C/5% $CO_2$, both utilizing polycarbonate Transwell inserts (8.0-µm-diameter pores, 6.5-mm-diameter, Costar cat. no. CLS3464-48EA). The remaining cells (9,000–80,000, depending on the experiment) were resuspended in 500 µl of serum-free D6 EVT medium and plated in the upper transwell insert chamber in duplicate. The lower chamber was filled with 500 µl of medium supplemented with 20% KSR. Cells were left to invade through the Matrigel for 48 or 72 h at 37 °C/5% $CO_2$. Non-migrated cells were removed from the upper chamber using a cotton bud, while cells that had migrated were fixed using 4% paraformaldehyde for 30 min and stained with 0.1% crystal violet solution for 15 min. Images were taken using a light microscope at ×10 magnification.

## Primary sequencing data processing
High-throughput sequencing reads (2 × 150-bp format; NovaSeq 6000) were trimmed using Trim_galore, with default settings. Mapping was done with either Bowtie2[88] (for CUT&Tag, CUT&RUN and ChIP-seq; default settings) or Hisat2[89] (for RNA-seq; with --no-softclip) to the reference genome of the species of origin: hg38, mm10 or rheMac10. Reads with MAPQ below 2 were discarded. Bigwig files were generated using deepTools[90] (bamCoverage tool with --binSize 200 --normalizeUsing CPM). Peak detection was performed using either MACS2[91] (with -q 0.05 --broad) or SEACR[92] (with 'norm' and 'relaxed' options). RNA-seq gene raw counts or reads per kilobase per million values were extracted using the RNA-seq pipeline in Seqmonk. DESeq2[93] (default parameters) was used to perform differential expression analysis. SQuIRE[94] was used to measure ERV family-wide expression from RNA-seq data.

## Peak enrichment at repeat families
For a given ChIP-seq/CUT&Tag/CUT&RUN experiment, the number of peaks overlapping each RepeatMasker-annotated repeat family was compared with overlap frequencies across 1,000 random controls (shuffled peaks, avoiding unmappable regions of the genome), yielding enrichment values and associated $P$ values. Significantly enriched repeat families had $P < 0.05$, >2-fold enrichment, and at least ten copies overlapped by peaks. Families were further selected by only keeping

those for which >80% of ENCODE placental DNase-seq samples had an enrichment above twofold, and a median enrichment difference larger than 2 when compared to liver, lung and kidney ENCODE DNase-seq data.

## Transcription factor motif analysis
Motifs enriched at active TE families were identified using the AME tool of the MEME suite[95] (default parameters) and the 2020 JASPAR vertebrate database. Relevant motifs were selected based on the expression of the respective transcription factors in hTSCs (>1 $\log_2$(FPKM)), followed by clustering to find redundant motifs and further selection based on literature searches. The FIMO tool was then used to extract motif locations and frequencies (default parameters).

## Human RNA-seq analysis
TE-derived promoters were identified by performing transcriptome assembly on primary cytotrophoblast data using Stringtie (with --rf and guided by the Gencode v38 annotation)[96], and intersecting the transcription start sites of multi-exonic transcripts with hTSC-active TE families. To evaluate putative enhancer effects, gene expression differences were determined based on the distance to the nearest H3K27ac-marked TE. JNK target genes were determined based on their distance to the nearest JUN binding peak. For all analyses, $\log_2$ fold differences were calculated using only genes that passed a minimal expression threshold (variable depending on dataset and normalization strategy) in at least one of the two samples compared. Gene ontology analysis of differentially expressed genes after JNK inhibition was performed using topGO.

## Comparative analysis
Human TE orthologs were identified in non-human primates by performing a reciprocal liftOver. Human TSC RNA-seq data were merged to either mouse TSC or rhesus macTSCs data based on one-to-one gene orthologs. For the human–mouse comparison, information about the proximity to hTSC-active TEs (from the selected families) and mTSC-active TEs (from the RLTR13D5 and RLTR13B families) was used to separate genes into different groups. For the human–macaque comparison, information about whether the nearest hTSC-active TE had a macaque ortholog was used.

## Statistical analyses
Wilcoxon tests with Benjamini-Hochberg correction for multiple comparisons or analysis of variance (ANOVA) followed by the Tukey post-hoc test were used, as specified in the figure legends.

## Reporting summary
Further information on research design is available in the Nature Portfolio Reporting Summary linked to this Article.

# Data availability
CUT&Tag, CUT&RUN, ChIP-seq and RNA-seq data have been deposited in NCBI's Gene Expression Omnibus under accession no. GSE200763. Details of other datasets used can be found in Supplementary Table 7. The GitHub repository https://github.com/MBrancoLab/Frost_2022_hTroph contains the following source data: (1) TE family enrichments for chromatin features ('Peak_enrichment' folder), (2) motif frequencies ('Transcription_factors/FIMO'), (3) JUN/JUND binding profiles ('Transcription_factors/AP1_profiles'), (4) processed RNA-seq data ('RNA-seq' and 'Comparative_analysis'), (5) TE orthology ('Active_families/orthologues') and (6) RT–qPCR, growth curves, FACS and ELISA ('Assays'). A Readme file is included describing how each figure was generated. Source data are provided with this paper.

# Code availability
All code associated with the manuscript is available in the GitHub repository https://github.com/MBrancoLab/Frost_2022_hTroph.

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

## Acknowledgements

We thank J. Kropp Schmidt and T. Golos for sharing their macaque TSC RNA-seq data, S. Henikoff (Fred Hutchinson Cancer Research Center), P. Madapura (QMUL), H. Wan (QMUL) and P. Hurd (QMUL) for reagents, and S. Papa (Univ. Leeds) for insights into the complexities of JNK activity. We also thank G. Warnes for his gracious assistance with FACS. This work was supported by the BBSRC (research grant no. BB/T000031/1, awarded to M.B. and J.F.), Barts Charity (MRC0297, awarded to M.B. and J.F.) and The Leakey Foundation (awarded to J.F.). This project has received funding from the European Union's Horizon 2020 research and innovation program under the Marie Skłodowska-Curie grant agreement InvADeRS no. 841172 (to J.F.).

## Author contributions

J.M.F. and M.R.B. designed the study and wrote the manuscript. J.M.F. performed cell culture, epigenomic, transcriptomic and genetic editing experiments in human cells. S.M.A. performed epigenomic experiments in mouse TSCs. H.O. and T.A. provided hTSCs and assisted in their culture. E.M.J. and M.P.C. performed immunofluorescence experiments. B.A., R.M.L. and J.K.C. isolated and cultured primary human cytotrophoblast. T.M. performed western blots and invasion assays. M.R.B. and J.M.F. performed bioinformatic analyses.

## Competing interests

The authors declare no competing interests.

## Additional information

**Extended data** is available for this paper at https://doi.org/10.1038/s41594-023-00960-6.

**Correspondence and requests for materials** should be addressed to Jennifer M. Frost or Miguel R. Branco.

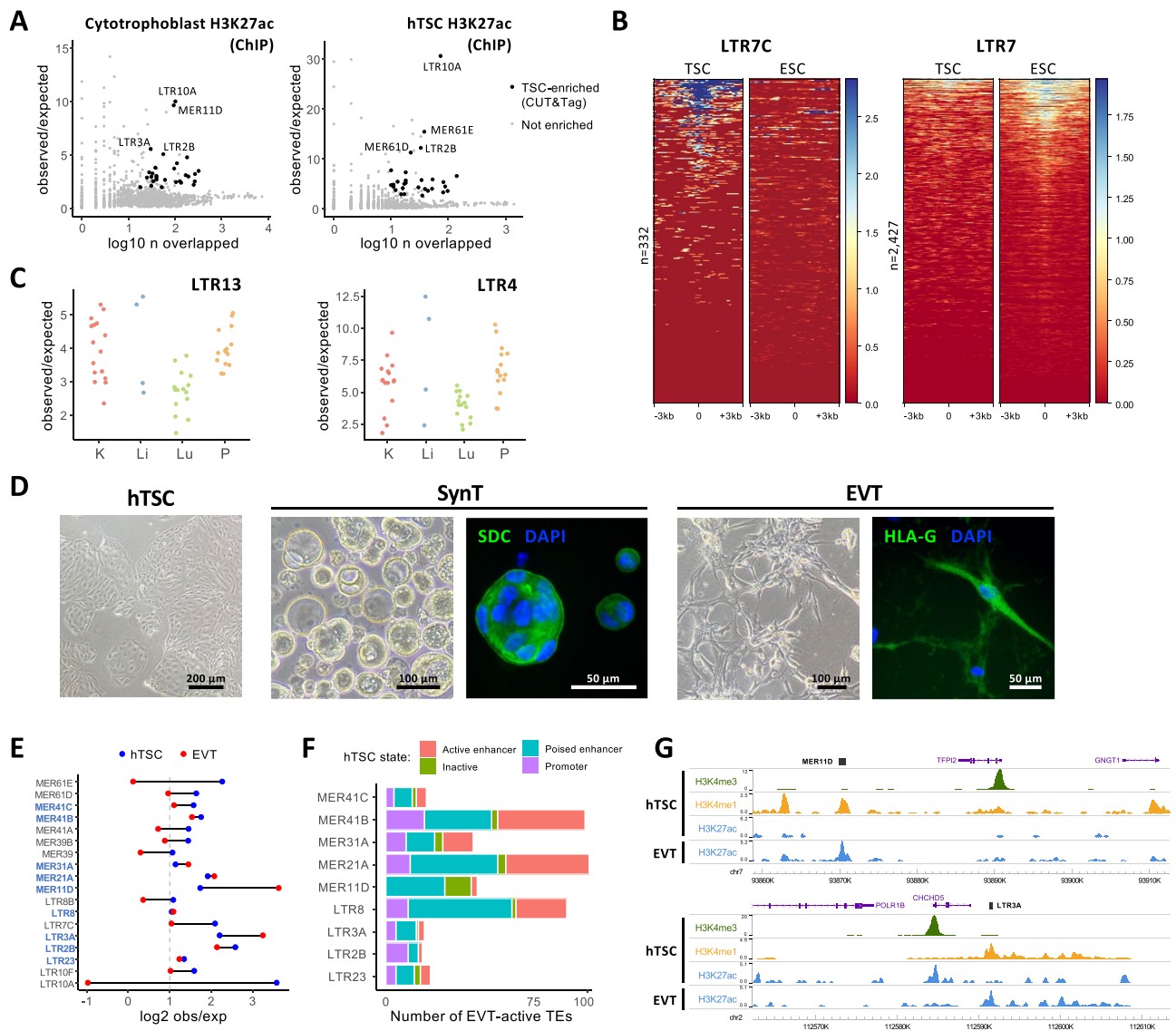

**Extended Data Fig. 1 | ERV regulatory signatures in undifferentiated and differentiated cells. A**, Enrichment for H3K27ac peaks for each repeat family in ChIP-seq data from primary cytotrophoblast ($n$ = 2) or hTSCs ($n$ = 1). Families with significant enrichment in CUT&Tag data from hTSCs (Fig. 1b) are highlighted. **B**, H3K27ac profiles of LTR7C and LTR7 families in hTSCs and hESCs. Each line represents an element in that family. **C**, Enrichment for Dnase hypersensitive sites in LTR13 and LTR4 families, in the kidney (K), liver (Li), lung (Lu) and placenta (P). Each datapoint represents a different ENCODE dataset. **D**, Representative phase contrast and immunofluorescence images for hTSCs and hTSC-derived SynT and EVT. Expression of the SynT marker SDC or EVT marker HLAG is shown (green), with DAPI counterstain (blue). **E**, Enrichment for H3K27ac peaks in hTSC or EVT for all hTSC-enriched ERV families. Families that are also significantly enriched in EVT are highlighted in blue. **F**, Number of H3K27ac-marked ERVs in EVT, divided by their chromatin state in hTSCs: active enhancer (H3K4me1 + H3K27ac), poised enhancer (H3K4me1 alone), promoter (H3K4me3) or inactive (none of the three marks). **G**, Genome browser snapshots showing examples of ERVs that are in a poised state in hTSCs and become active in EVT. Please see the data availability statement for details on source data for panels **A**, **C**, **E** and **F**.

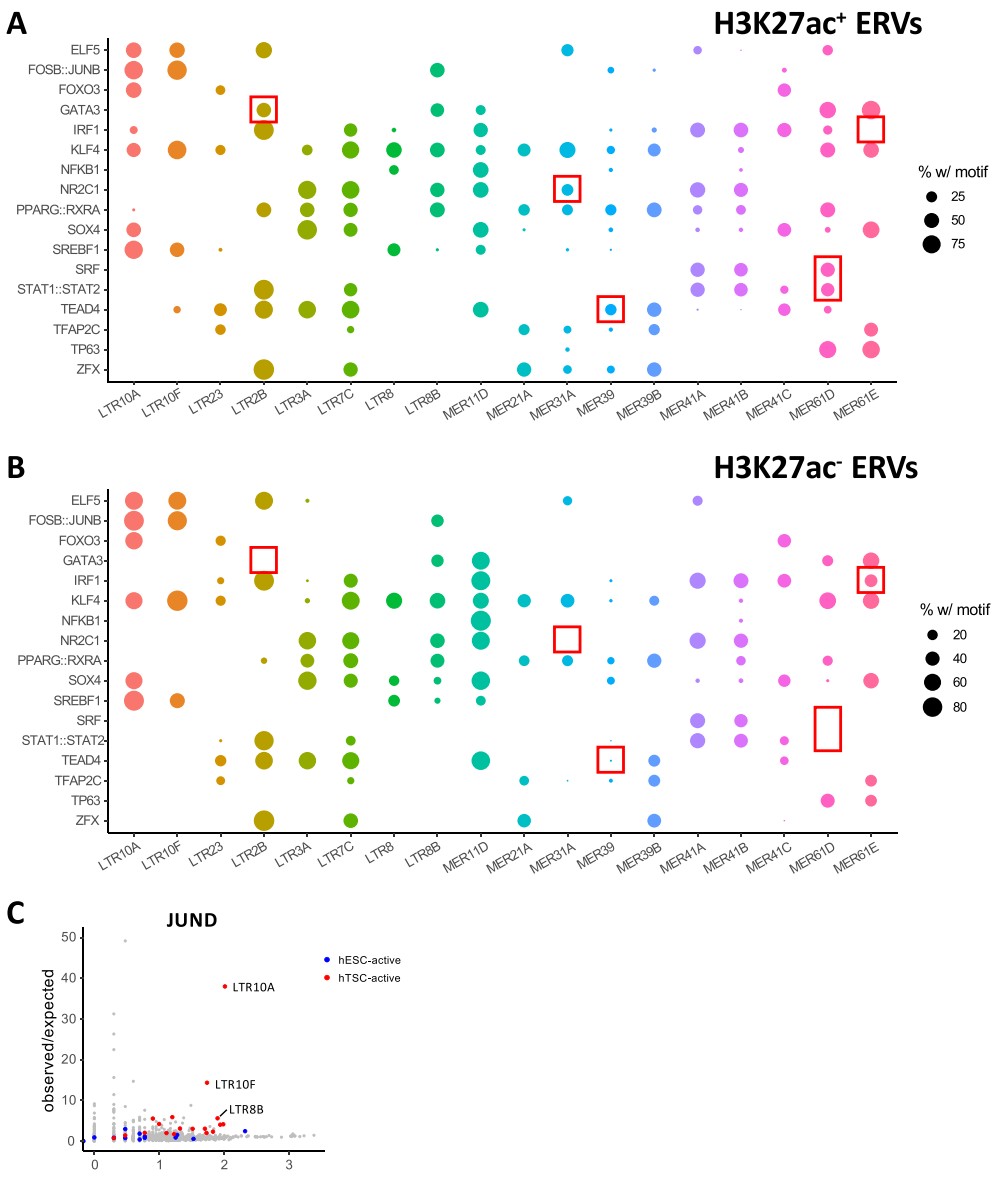

**Extended Data Fig. 2 | Transcription factor binding motifs for all hTSC-active ERV families. A**, Proportion of H3K27ac-marked elements from all analyzed ERV families bearing motifs for the transcription factors on the y axis. **B**, As in **B**, but for H3K27ac-negative elements. Particularly striking examples of differential motif enrichment between H3K27ac+ and H3K27ac- ERVs are highlighted. **C**, Repeat family-wide enrichment for peaks from JUND CUT&Tag data on hTSCs. H3K27ac-enriched families in hTSCs or hESCs are highlighted. Please see the data availability statement for details on source data.

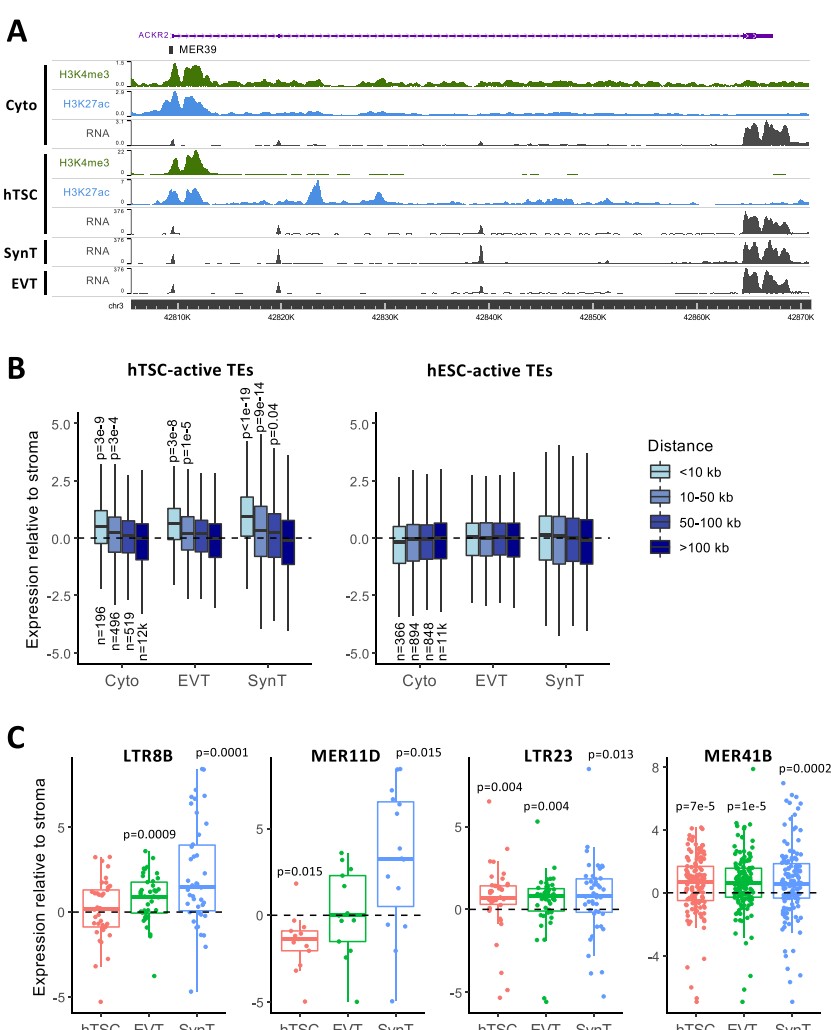

**Extended Data Fig. 3 | Associations between hTSC-active ERVs and gene expression. A**, Genome browser snapshot showing an example of an ERV-derived gene promoter in primary cytotrophoblast. **B**, Gene expression in primary cytotrophoblast, EVT and SynT relative to placental stroma. Genes are grouped based on their distance to the nearest H3K27ac-marked TE in hTSCs. *P* values are for the difference to the '>100 kb' group based on ANOVA and Tukey post-hoc test. **c**) Expression in hTSCs and hTSC-derived EVT and SynT relative to primary placental stroma for genes within 50 kb of an H3K27ac-marked ERV of the indicated family. Genes are grouped based on their distance to the nearest H3K27ac-marked TE in hTSCs. *P* values are from two-sided Wilcoxon tests comparing each distribution to 0, with multiple comparisons correction. Boxplots show median center, 25th and 75th percentile box bounds, and 1.5× IQR whisker limits. Please see the data availability statement for details on source data for panels **B** and **C**.

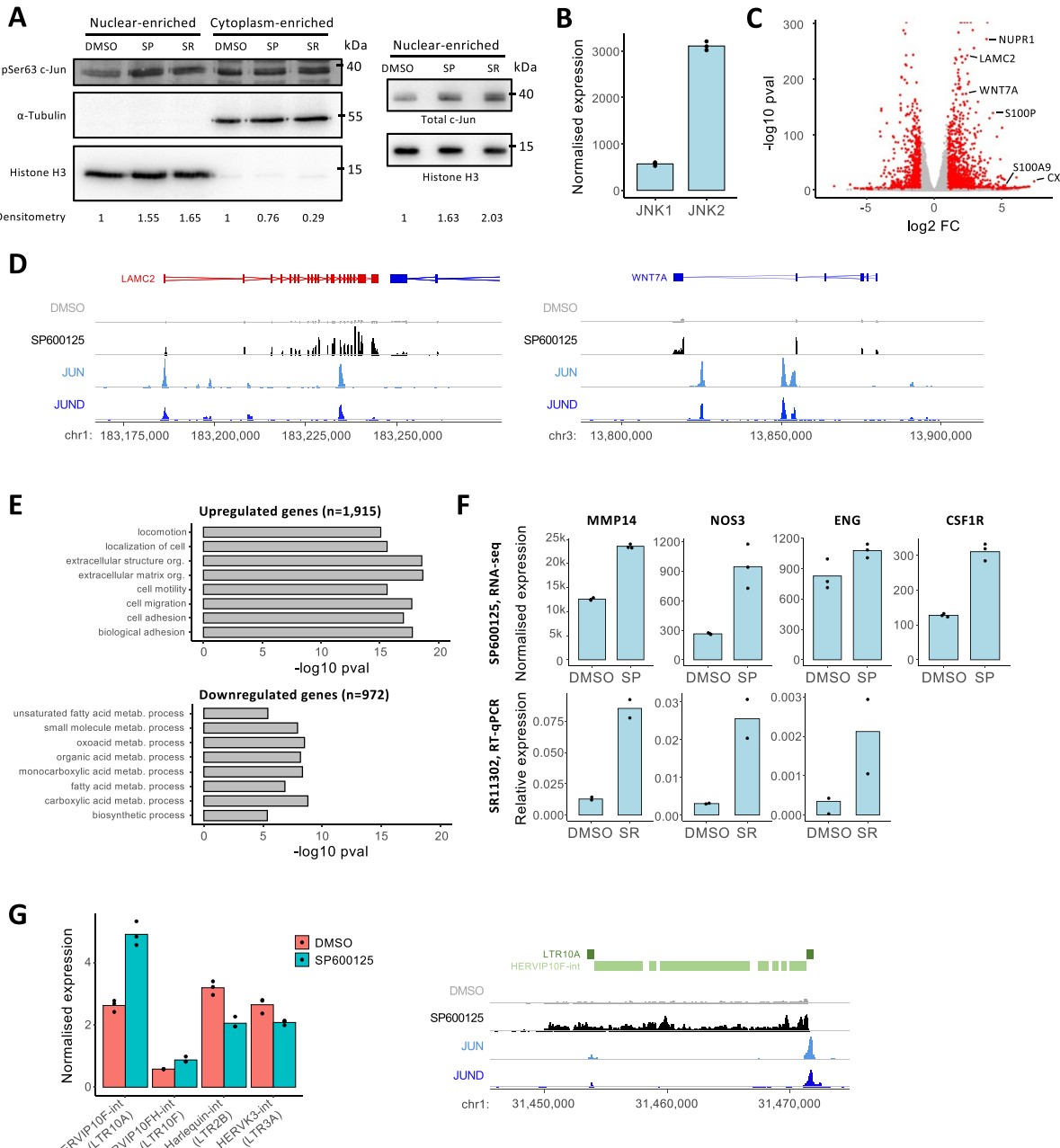

**Extended Data Fig. 4 | Effects of SP600125 treatment in hTSCs. A**, Western blot of phosphorylated c-Jun (pSer63 c-Jun) and total c-Jun in enriched nuclear or cytoplasmic fractions of cells treated with the AP-1 inhibitors SP600125 ('SP') or SR11302 ('SR'). Densitometry values normalized to the respective loading controls are shown below. Full blot images are in Supplementary Figure 7. **B**, Expression of *JNK1* (*MAPK8*) and *JNK2* (*MAPK9*) in hTSCs (data from RNA-seq). **C**, Volcano plot of RNA-seq data from cells treated with SP600125. *P* values are from the DESeq2 R package. **D**, Genome browser snapshots showing examples of genes highlighted in C, together with CUT&Tag data for JUN and JUND. **E** Gene ontology biological processes enriched terms for genes up- or downregulated after treatment of hTSCs with SP600125. *P* values are from the topGO R package. **F**, Expression of *MMP14* (a known AP-1 target), *NOS3*, *ENG* and *CSF1R* in hTSCs treated with SP600125 (data from RNA-seq) and/or SR11302 (see data from RT–qPCR). **G**, Expression of internal ERV fragments driven by LTRs of H3K27ac-enriched families (in brackets) upon SP600125 treatment. An example of LTR10A-driven transcription of a proviral HERVIP10-F locus is shown on the right. Please see the data availability statement for details on source data for panels **B**, **C** and **E–G**.

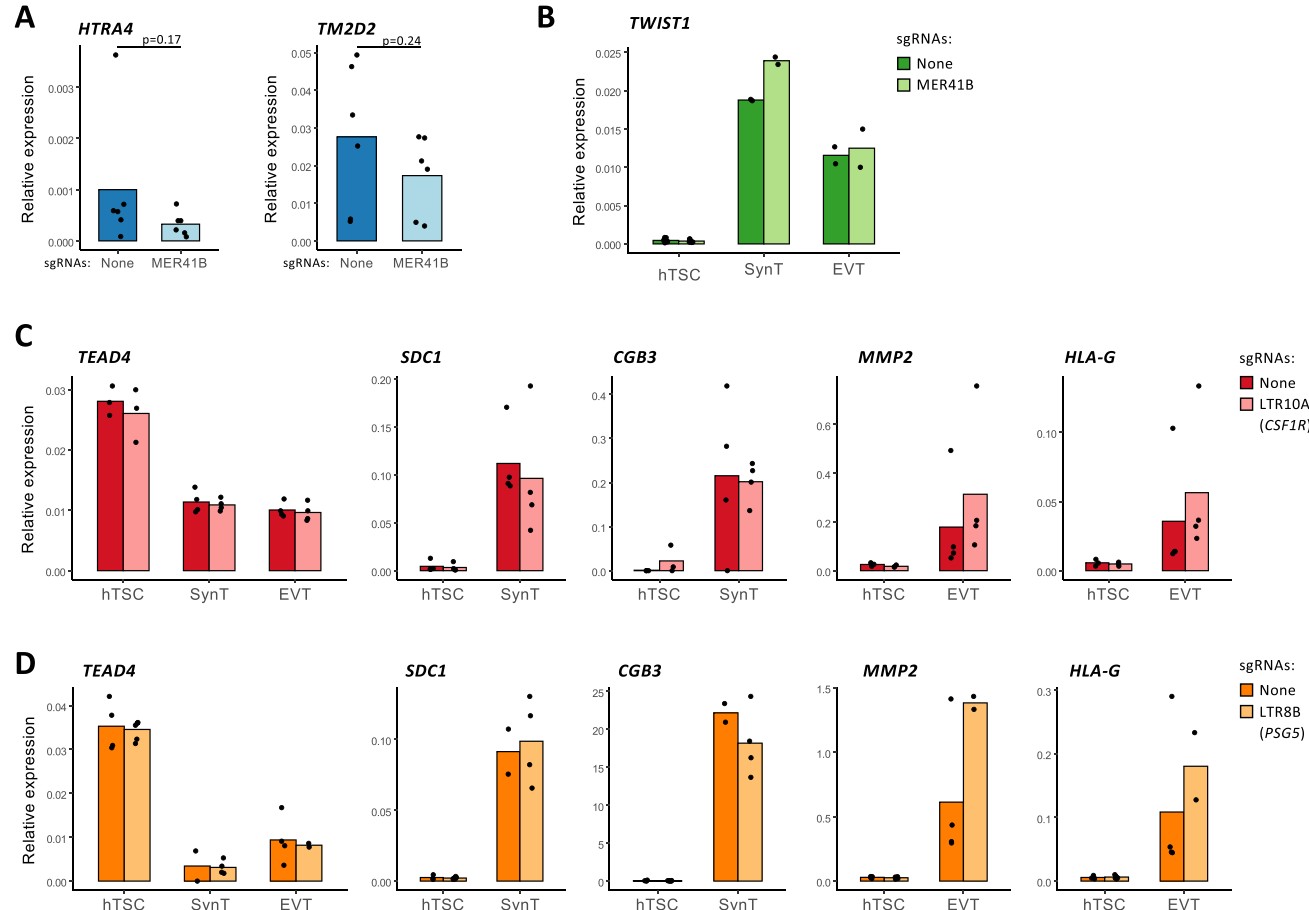

**Extended Data Fig. 5 | Effects of ERV genetic excisions on other genes.**
**A**, RT–qPCR data for *HTRA4* and *TM2D2* in hTSC populations treated with lentiviral CRISPR constructs carrying either no sgRNAs or sgRNAs that excise the MER41B element highlighted in Fig. 5a (*n* = 6). **B**, *TWIST1* expression in EVT and SynT derived from hTSC populations with no sgRNAs or sgRNAs that excise the MER41B element highlighted in Fig. 5c. **C**, Expression of stem cell (*TEAD4*), SynT (*SDC1*, *CGB3*) and EVT (*MMP2*, *HLAG*) markers in cell populations with no sgRNAs or sgRNAs that excise the LTR10A element highlighted in Fig. 5b. **D**, As in **C**, but for the LTR8B element in Fig. 5d. No significant differences were detected in any of the data after two-sided Wilcoxon tests with multiple comparisons correction. Please see the data availability statement for details on source data.

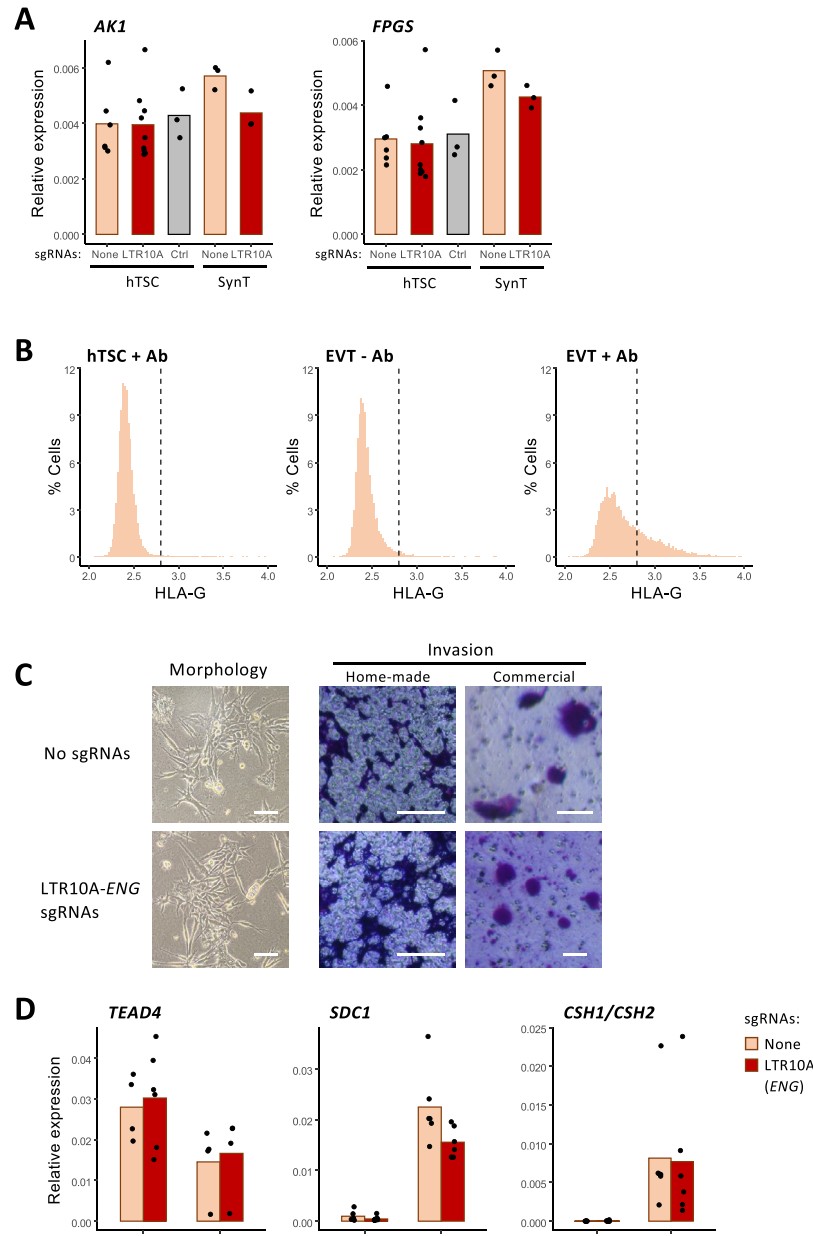

**Extended Data Fig. 6 | Transcriptional and phenotypic effects of LTR10A-*ENG* deletion. A**, *AK1* and *FPGS* expression in hTSC or hTSC-derived SynT populations carrying lentiviral CRISPR constructs with no sgRNAs, sgRNAs that excise the LTR10A element highlighted in Fig. 6a, or sgRNAs that excise a control region also highlighted in Fig. 6a. **B,** Representative FACS profiles for HLAG immunolabelling in hTSCs and hTSC-derived EVTs. The middle panel is a no-antibody control. **C**, Images of cells in culture (to assess morphology) or crystal violet-stained cells after an invasion assay using Matrigel-coated chambers (either home-made chambers or from a commercial provider). Images representative from six independent experiments. Scale bars are 100 μm. **D**, Expression of stem cell (*TEAD4*) and SynT (*SDC1*, *CSH1/CSH2)* in cell populations with no sgRNAs or sgRNAs that excise the LTR10A-*ENG* element. No significant differences were detected in any of the qRT–PCR data after Wilcoxon tests with multiple comparisons correction. Please see the data availability statement for details on source data for panels **A**, **B** and **D**.

# Reporting Summary

## Statistics

For all statistical analyses, confirm that the following items are present in the figure legend, table legend, main text, or Methods section.

| n/a | Confirmed | |
|---|---|---|
| ☐ | ☒ | The exact sample size (*n*) for each experimental group/condition, given as a discrete number and unit of measurement |
| ☐ | ☒ | A statement on whether measurements were taken from distinct samples or whether the same sample was measured repeatedly |
| ☐ | ☒ | The statistical test(s) used AND whether they are one- or two-sided *Only common tests should be described solely by name; describe more complex techniques in the Methods section.* |
| ☒ | ☐ | A description of all covariates tested |
| ☐ | ☒ | A description of any assumptions or corrections, such as tests of normality and adjustment for multiple comparisons |
| ☐ | ☒ | A full description of the statistical parameters including central tendency (e.g. means) or other basic estimates (e.g. regression coefficient) AND variation (e.g. standard deviation) or associated estimates of uncertainty (e.g. confidence intervals) |
| ☐ | ☒ | For null hypothesis testing, the test statistic (e.g. *F*, *t*, *r*) with confidence intervals, effect sizes, degrees of freedom and *P* value noted *Give P values as exact values whenever suitable.* |
| ☒ | ☐ | For Bayesian analysis, information on the choice of priors and Markov chain Monte Carlo settings |
| ☒ | ☐ | For hierarchical and complex designs, identification of the appropriate level for tests and full reporting of outcomes |
| ☒ | ☐ | Estimates of effect sizes (e.g. Cohen's *d*, Pearson's *r*), indicating how they were calculated |

*Our web collection on statistics for biologists contains articles on many of the points above.*

## Software and code

Policy information about availability of computer code

| Data collection | No software was used for data collection. |
|---|---|
| Data analysis | Open source software: trim galore, Bowtie2 v2.1.0, MACS2 v2.1.1 , SEACR v1.2 , Hisat2 v2.0.5 , DESeq2 v1.36.0, deepTools2.0, StringTie v1.3.3b, MEME SUITE v5.0.1, Seqmonk v1.47.2, SQuIRE v0.9.9.9a-beta. Custom scripts used for data analysis are available at https://github.com/MBrancoLab/Frost_2022_hTroph. |

For manuscripts utilizing custom algorithms or software that are central to the research but not yet described in published literature, software must be made available to editors and reviewers. We strongly encourage code deposition in a community repository (e.g. GitHub). See the Nature Portfolio guidelines for submitting code & software for further information.

## Data

Policy information about availability of data

All manuscripts must include a data availability statement. This statement should provide the following information, where applicable:
- Accession codes, unique identifiers, or web links for publicly available datasets
- A description of any restrictions on data availability
- For clinical datasets or third party data, please ensure that the statement adheres to our policy

CUT&Tag, CUT&RUN, ChIP-seq and RNA-seq data have been deposited in NCBI's Gene Expression Omnibus under accession number GSE200763.
Details of other datasets used can be found in Supplementary Table S7.
Genome assemblies used: hg38, mm10, calJac4, gorGor6, micMur2, nomLeu3, panTro6, ponAbe3, rheMac10, tarSyr2.
The Github repository https://github.com/MBrancoLab/Frost_2022_hTroph contains the following source data: 1) TE family enrichments for chromatin features ('Peak_enrichment' folder), 2) motif frequencies ('Transcription_factors/FIMO'), 3) JUN/JUND binding profiles ('Transcription_factors/AP1_profiles'), 4) processed

RNA-seq data ('RNA-seq' and 'Comparative_analysis'), 5) TE orthology ('Active_families/orthologues'), 6) RT-qPCR, growth curves, FACS and ELISA ('Assays'). A Readme file is included describing how each figure was generated.

# Field-specific reporting

Please select the one below that is the best fit for your research. If you are not sure, read the appropriate sections before making your selection.

☒ Life sciences      ☐ Behavioural & social sciences      ☐ Ecological, evolutionary & environmental sciences

For a reference copy of the document with all sections, see nature.com/documents/nr-reporting-summary-flat.pdf

# Life sciences study design

All studies must disclose on these points even when the disclosure is negative.

| | |
|---|---|
| Sample size | Sample sizes were not statistically predetermined, but were based on similar studies. TE-derived enhancers were derived from two replicates in hTSCs, with similar results in another two independent datasets (see 'Replication' below). RNA-seq of SP600125-treated cells was performed in triplicate and sufficiently powered to detect 10% differences at p<0.05. The effects of CRISPR experiments were performed in variable numbers of replicates (see details in manuscript) depending on the assay and observed variability; effect sizes of ~2-fold could be robustly detected. |
| Data exclusions | No data were excluded from the analyses. |
| Replication | TE-derived enhancers in human trophoblast were derived from two hTSC CUT&Tag runs, one hTSC ChIP-seq and two published ChIP-seq sets from primary trophoblast, with similar results. All results from RNA-seq data analyses were successfully replicated between hTSCs and primary cytotrophoblast. Results from CRISPR experiments were successfully replicated by performing independent infections and/or using different sgRNA pairs (as detailed in the main text and figure legend). |
| Randomization | This study used cell lines, for which randomization is not applicable. When treatments were used (CRISPSR, JNK inhibition), the same cell population was split into two or more equal subpopulations. |
| Blinding | Blinding was used for manually quantifying cell invasion assays. All other data analyses were independent of subjective human-based decisions, using objective and standardised data analysis pipelines, and therefore blinding wasn't used. |

# Reporting for specific materials, systems and methods

We require information from authors about some types of materials, experimental systems and methods used in many studies. Here, indicate whether each material, system or method listed is relevant to your study. If you are not sure if a list item applies to your research, read the appropriate section before selecting a response.

## Materials & experimental systems

| n/a | Involved in the study |
|---|---|
| ☐ | ☒ Antibodies |
| ☐ | ☒ Eukaryotic cell lines |
| ☒ | ☐ Palaeontology and archaeology |
| ☒ | ☐ Animals and other organisms |
| ☒ | ☐ Human research participants |
| ☒ | ☐ Clinical data |
| ☒ | ☐ Dual use research of concern |

## Methods

| n/a | Involved in the study |
|---|---|
| ☐ | ☒ ChIP-seq |
| ☒ | ☐ Flow cytometry |
| ☒ | ☐ MRI-based neuroimaging |

## Antibodies

| | |
|---|---|
| Antibodies used | HLAG (APC anti-human HLA-G Antibody Clone: 87G, BioLegend UK #335905)<br>H3K4me3 (RRID:AB_2616052, Diagenode C15410003)<br>H3K4me1 (RRID:AB_306847, Abcam ab8895)<br>H3K27ac (RRID:AB_2637079, Diagenode C15410196)<br>H3K27Ac (39034, Active Motif)<br>H3K9me3 (C15410193, Diagenode)<br>H3K27me3 (C15410195, Diagenode)<br>c-Jun (60A8 – 9165T, Cell Signalling)<br>JunD (D17G2 – 5000S, Cell Signalling)<br>GATA3 (sc-268, Santa Cruz)<br>TFAP2C (sc-12762, Santa Cruz)<br>TEAD4 (CSB-PA618010LA01HU, Stratech)<br>rabbit IgG (sc-2027, Santa Cruz)<br>Guinea Pig anti-Rabbit IgG (ABIN101961, Antibodies-Online) |

pSer63 c-Jun (#2361, Cell Signalling)
α-Tubulin (T9026, Sigma Aldrich)
H3 (ab1791, Abcam)
peroxidase conjugated anti-rabbit IgG  (A6154, Sigma Aldrich)
peroxidase conjugated anti-mouse IgG (A0168, Sigma Aldrich)
SDC-1/CD138 (clone MI15, Fisher Scientific #15892669)
Goat anti-Mouse IgG, Alexa Fluor Plus 488 (Catalog # A32723, Thermo Fisher Scientific)

| Validation | HLAG - clone 87G has been validated by HLA-G overexpression in PMID:34201301, and used for IF and flow cytometry. |
|---|---|
| | ChIP-grade antibodies from Diagenode (H3K4me3, H3K27ac, H3K9me3, H3K27me3) were extensively validated by the manufacturer using dot blots, western blots, ChIP-seq, and others. |
| | H3K4me1 (Abcam ab8895, ChIP-grade) - validated by western blot and ChIP-seq (manufacturer's website). |
| | H3K27ac (Active Motif 39034) - validated by the manufacturer for ChIP-seq (and similar), western blot and IF. |
| | c-Jun (60A8 – 9165T, Cell Signalling) - validated for western blot using c-Jun knockout HeLa cells (manufacturer's website). |
| | JunD (D17G2 – 5000S, Cell Signalling) - validated in PMID:35926467 by JunD knockdown. |
| | Santa Cruz antibodies (GATA3, TFAP2C) were validated by overexpression of the respective protein (manufacturer's website). |
| | TEAD4 (CSB-PA618010LA01HU, Stratech) - validated by western blot (Cusabio website). |
| | pSer63 c-Jun (#2361, Cell Signalling) - validated by activation of c-Jun kinases using anisomycin (manufacturer's website). |
| | α-Tubulin (T9026, Sigma Aldrich) - validated by western blot and IF (manufacturer's website). |
| | H3 (ab1791, Abcam) - validated by the manufacturer by blocking the antibody with an H3 peptide. |
| | SDC-1/CD138 (clone MI15) - validated in PMID:10027728 using recombinant proteins. |

# Eukaryotic cell lines

Policy information about cell lines

| Cell line source(s) | hTSCs were obtained from the RIKEN cell bank (RIKEN BioResource Research Center, 305-0074 Japan). HEK293T cells were a gift from Prof. Silvia Marino (QMUL) |
|---|---|
| Authentication | hTSCs were validated by qRT-PCR of marker genes, differentiation into ST and EVT. No validation was performed on 293T cells, as they were used just to produce lentiviruses. |
| Mycoplasma contamination | Cells tested negative for mycoplasma |
| Commonly misidentified lines (See ICLAC register) | No commonly misidentified cell lines were used. |

# ChIP-seq

## Data deposition

☒ Confirm that both raw and final processed data have been deposited in a public database such as GEO.

☒ Confirm that you have deposited or provided access to graph files (e.g. BED files) for the called peaks.

| Data access links *May remain private before publication.* | https://www.ncbi.nlm.nih.gov/geo/query/acc.cgi?acc=GSE200763 |
|---|---|
| Files in database submission | Raw fastq files, peak files, bigwig tracks. |
| Genome browser session (e.g. UCSC) | Human: http://epigenomegateway.wustl.edu/browser/?sessionFile=https://data.cyverse.org/dav-anon/iplant/home/mbranco/hTroph/eg-session-lV_cZJGeY-ec032020-d6e7-11ec-9604-a1ccb72a2b23.json Mouse: http://epigenomegateway.wustl.edu/browser/?sessionFile=https://data.cyverse.org/dav-anon/iplant/home/mbranco/mTroph/eg-session--41d93570-d6ed-11ec-a762-1750142ba3bb.json |

## Methodology

| Replicates | hTSC and mTSC H3K27ac, H3K4me1 and H3K4me3 CUT&Tag were performed in duplicate. All other experiments are single replicates. |
|---|---|

| Sequencing depth | Sample: | Read count: |
|---|---|---|
| | hTSC H3K27ac ChIP | 40,213,296 |
| | hTSC H3K4me1 ChIP | 79,194,135 |
| | hTSC H3K4me3 ChIP | 61,449,294 |
| | hTSC Input ChIP | 83,823,113 |
| | hTSC H3K27ac CUT&Tag 1 | 7,871,525 |
| | hTSC H3K27ac CUT&Tag 2 | 4,859,297 |
| | hTSC H3K4me1 CUT&Tag 1 | 8,861,507 |
| | hTSC H3K4me1 CUT&Tag 2 | 12,678,027 |
| | hTSC H3K4me3 CUT&Tag 1 | 7,867,045 |
| | hTSC H3K4me3 CUT&Tag 2 | 19,393,419 |
| | hTSC cJun CUT&Tag | 5,652,603 |
| | hTSC JunD CUT&Tag | 5,864,304 |
| | hTSC IgG CUT&Tag | 2,486,061 |

| hTSC H3K9me3 CUT&Tag | 19,549,401 |
|---|---|
| hTSC H3K27me3 CUT&Tag | 6,095,557 |
| hTSC GATA3 CUT&RUN | 3,194,528 |
| hTSC TEAD4 CUT&RUN | 21,992,284 |
| hTSC TFAP2C CUT&RUN | 5,407,256 |
| hTSC IgG CUT&RUN | 3,538,560 |
| EVT H3K27ac CUT&Tag | 3,892,494 |
| EVT IgG CUT&Tag | 1,226,230 |
| mTSC H3K27ac CUT&Tag 1 | 16,719,388 |
| mTSC H3K27ac CUT&Tag 2 | 9,610,089 |
| mTSC H3K4me1 CUT&Tag 1 | 15,974,036 |
| mTSC H3K4me1 CUT&Tag 2 | 16,650,467 |
| mTSC H3K4me3 CUT&Tag 1 | 16,803,789 |
| mTSC H3K4me3 CUT&Tag 2 | 10,955,327 |
| mTSC IgG CUT&Tag 1 | 4,583,747 |
| mTSC IgG CUT&Tag 2 | 567,885 |

**Antibodies**

H3K4me3 (RRID:AB_2616052, Diagenode C15410003)
H3K4me1 (RRID:AB_306847, Abcam ab8895)
H3K27ac (RRID:AB_2637079, Diagenode C15410196)
H3K27Ac (39034, Active Motif)
H3K9me3 (C15410193, Diagenode)
H3K27me3 (C15410195, Diagenode)
c-Jun (60A8 – 9165T, Cell Signalling)
JunD (D17G2 – 5000S, Cell Signalling)
GATA3 (sc-268, Santa Cruz)
TFAP2C (sc-12762, Santa Cruz)
TEAD4 (CSB-PA618010LA01HU, Stratech)
rabbit IgG (sc-2027, Santa Cruz)
Guinea Pig anti-Rabbit IgG (ABIN101961)

**Peak calling parameters**

ChIP-seq peaks were called using MACS2 v2.1.1 with -q 0.05 and --broad.
CUT&Tag and CUT&RUN peaks were called using SEACR v1.2 with normalisation to IgG and relaxed mode.

**Data quality**

FastQC was used to assess raw data quality. Peak detection was performed with a q-value cut-off of 0.05.

**Software**

MACS2 v2.1.1 and SEACR v1.2 for peak detection. Other custom scripts are available at https://github.com/MBrancoLab/Frost_2022_hTroph.

