## [Peer Review File · Nature Structural & Molecular Biology]

Peer Review Information

Manuscript Title: Regulation of human trophoblast gene expression by endogenous retroviruses

Corresponding author name(s): Jennifer Frost, Miguel Branco

Reviewer Comments & Decisions:

Decision Letter, initial version:

Message: 10th Aug 2022

Dear Dr. Branco,

Thank you again for submitting your manuscript "Regulation of human trophoblast gene expression by endogenous retroviruses". I apologize for the delay in responding, which resulted from the difficulty in obtaining suitable referee reports. Nevertheless, we now have comments (below) from the 3 reviewers who evaluated your paper. In light of those reports, we remain interested in your study and would like to see your response to the comments of the referees, in the form of a revised manuscript.

You will see that while the referees appreciate the study and the insights it provides for potential role of ERVs as lineage-specific placental enhancers, Reviewer #3 suggests that more mechanistic data is needed to understand the role of ERVs acting as enhancers. We agree that this additional analysis would strengthen the manuscript and increase its impact. Please be sure to also respond to all other technical concerns of the referees in full in a point-by-point response and highlight all changes in the revised manuscript text file. If you have comments that are intended for editors only, please include those in a separate cover letter.

We are committed to providing a fair and constructive peer-review process. Please do reach out if there are specific requests from the reviewers that you believe are technically impossible or unlikely to yield a meaningful outcome.

We usually expect to see a revised manuscript within 6 weeks, but we do understand that revisions may take longer this time of year. If you cannot send it within this time, please contact us to discuss an extension; we would still consider your revision, provided that no similar work has been accepted for publication at NSMB or published elsewhere.

Reporting Summary:

When submitting the revised version of your manuscript, please pay close attention to our [href="https://www.nature.com/nature-portfolio/editorial-policies/image-integrity">Digital Image Integrity Guidelines. and to the following points below:](https://www.nature.com/nature-portfolio/editorial-policies/image-integrity)

Please note that all key data shown in the main figures as cropped gels or blots should be presented in uncropped form, with molecular weight markers. These data can be aggregated into a single supplementary figure item. While these data can be displayed in a relatively informal style, they must refer back to the relevant figures. These data should be submitted with the final revision, as source data, prior to acceptance, but you may want to start putting it together at this point.

Data availability: this journal strongly supports public availability of data. All data used in accepted papers should be available via a public data repository, or alternatively, as

Supplementary Information. If data can only be shared on request, please explain why in your Data Availability Statement, and also in the correspondence with your editor. Please note that for some data types, deposition in a public repository is mandatory - more information on our data deposition policies and available repositories can be found below: <https://www.nature.com/nature-research/editorial-policies/reporting-standards#availability-of-data>

We require deposition of coordinates (and, in the case of crystal structures, structure factors) into the Protein Data Bank with the designation of immediate release upon publication (HPUB). Electron microscopy-derived density maps and coordinate data must be deposited in EMDb and released upon publication. Deposition and immediate release of NMR chemical shift assignments are highly encouraged. Deposition of deep sequencing and microarray data is mandatory, and the datasets must be released prior to or upon publication. To avoid delays in publication, dataset accession numbers must be supplied with the final accepted manuscript and appropriate release dates must be indicated at the galley proof stage.

Nature Structural & Molecular Biology is committed to improving transparency in authorship. As part of our efforts in this direction, we are now requesting that all authors identified as 'corresponding author' on published papers create and link their Open Researcher and Contributor Identifier (ORCID) with their account on the Manuscript Tracking System (MTS), prior to acceptance. This applies to primary research papers only. ORCID helps the scientific community achieve unambiguous attribution of all scholarly contributions. You can create and link your ORCID from the home page of the MTS by clicking on 'Modify my Springer Nature account'. For more information please visit [visit www.springernature.com/orcid](http://www.springernature.com/orcid).

[Redacted]

Sincerely,

Carolina

Carolina Perdigoto, PhD
Chief Editor
Nature Structural & Molecular Biology

orcid.org/0000-0002-5783-7106

Referee expertise:

Referee #1: trophoblast lineage, placenta development, reproduction, TEs

Referee #2: TEs, epigenetics, stem cells and development

Referee #3: TEs, bioinformatics

Reviewers' Comments:

Reviewer #1:

Remarks to the Author:

Summary

This paper is of interest to readers in the fields of epigenetics, genome biology, evolutionary biology, genetics, placental biology and transposable elements/endogenous retroviruses (ERVs). Using recently established human trophoblast stem cells and differentiated extravillous trophoblast and syncytiotrophoblast cells, the authors identify several ERV families and individual ERV elements with species- and placenta-specific regulatory potential. The authors use a clever screening strategy to identify putative enhancer and promoters by combining histone "marks" and placental transcription factor binding at ERV subfamilies with the tissue- and species-specific expression of nearby genes. The authors propose that these regulatory elements are important for placental- and species-specific expression of nearby genes, which they substantiate with genes nearby human-specific elements showing lower expression in hESCs or macaque and mouse TSCs than in hTSCs. Population-based CRISPR/Cas9-based gene editing which leads to a reduction in the presence of six individual ERV-derived regulatory elements in pooled undifferentiated and differentiated hTSCs support this main claim by showing altered expression of the nearby genes. Sometimes this change is seen in hTSCs and sometimes only upon differentiation into extra-villous trophoblast or syncytiotrophoblast depending on the specifics of each LTR element. For one identified ERV-regulated gene, ENG, the authors also find reduced protein levels in placental cells. While this increase seems not to affect hTSC differentiation, an increase of H3K27Ac over this element specifically in pre-eclampsia human placenta indicates a potential role for the ERV-derived enhancer in preeclampsia. The experiments are well-controlled, the presentation of the data is clear, and the manuscript is well written. This work is a substantial step forward for the field in its aim to test how species-specific ERVs may have shaped human placental development and function.

Strengths:

- Given the constraints of working in human models, the authors do an excellent job exploiting recently developed human placental stem cell models. The characterization of these new models in respect to putative human-specific regulatory elements derived from TEs, which have been shown to provide species-specific regulatory elements in mice and primates, is a needed and relevant analysis to understand gene regulation in human

pregnancy and pregnancy complication such as preeclampsia.

- The authors data is convincing because the experiments are well-controlled. For example, H3K27ac-enriched TE families are identified by screening both primary cytotrophoblast and an independently established novel cytotrophoblast-derived hTSC line and only overlapping peaks are considered.
- Demonstrating the feasibility of gene editing to reduce individual elements of individual TE subfamilies in pools of hTSC and differentiated EVT and SynT cells derived from hTSCs is a technological advance important for the human placental biology field. Trying out the population-wide lentiviral approach used by the authors before in mice in this human system is applauded and supports a functional role of the ERVs who are partially eliminated in these experiments leading in 4/6 cases tested to altered expression of the nearby gene. The authors also went beyond the TSC stage and examined expression changes upon differentiation, allowing them to discover important ERV derived regulatory elements, including the LTR8B-driven PSG5 gene and LTR10A driven CSFR1 gene where the effect of the ERV on gene expression becomes apparent upon differentiation. The possible effect of an ENG enhancer on ENG protein levels is intriguing, even though this increase seems to not functionally affect hTSC differentiation. The finding of increased H3K27ac ChIP-seq signal in preeclamptic pregnancies at some of the specific LTR elements is potentially quite important, indicating that genetic or epigenetic changes at ERVs might contribute to maternal complications during pregnancy, and I like that the authors mention implications of analyzed genes for preeclampsia throughout the text.
- Using hTSCs differentiated to extra-villous trophoblast the authors demonstrate that, for some repeat subfamilies, a third to half of H3K4me1-only marked regions in hTSCs gain H3K27ac in EVTs in agreement with them being poised enhancers that become active upon differentiation.
- The authors not only identify TF motifs but show the gain/loss of TF binding at individual ERV subfamilies, such as in Fig 2D. They provide compelling evidence that specific TE subfamilies can evolve/provide binding sites for TFs (potentially in order to remain expressed after endogenization).
- By exploiting available mouse and macaque datasets and comparison of putative ERV-regulated human genes the authors strengthen the proposed role of ERVs in shaping human-specific gene regulatory networks. The few convergently evolved ERV-driven genes in mouse and human identified (although only 1 is shown) are intriguing!

Weaknesses:

- the authors do not discuss if there is anything unique about the repeat families co-opted in placenta vs hESCs, which may be relevant given that placental morphology is generally considered to have evolved more rapidly than embryos, and as the authors point out that TEs may fuel fast placental evolution across primates (in Fig4A and lines 299.) While an analysis of orthologous regions (evolutionary conservation) is done in Fig4B for primate species, it would be useful to assess ages of TEs coopted for hTSCs vs hESCs. Are the cooped TEs in TSC/ESC younger/older? less conserved/similarly conserved? more ERVs than other TE families or similar between ESC/TSC? for example: what distinguishes the coopted LTR7up hESC and LTR7C hTSC coopted TEs? Age? Sequence?
- Line 131: 'more specifically, it is the LTR7up subfamily that is active is hESCs' - Figure S1A does only show LTR7 elements: was it challenging to display LTR7up subfamily elements or why is this statement included in the main text here but not represented in the data included here (LTR7up must be contained in the LTR7 peaks?) If possible, the authors could add a subset heatmap with the LTR7up repeat elements?
- line 172f: 'focused on a selection of transcription factors that are expressed in

trophoblast' How were these transcription factors and their expression identified (expression cut-off or just picked examples of known placentally expressed TFs)?

- Line 206-212: 'active elements' I suspect this means marked by certain marks (H3K4me1/H3K27ac) in hTSCs? Please clarify what active means (e.g. marked by H3K4me1/H3K27ac in hTSCs)

- line 283: judging from Fig 2C both LTR10A and LTR10F are enriched for JUN binding so I would expect target genes of both subfamilies to contribute to observed gene expression changes upon JNK inhibition. Can the authors speculate why only target genes of one change?

Additional Recommendations to Authors:

- line 201/202: the authors reference Fig 2C and S2B in which, unlike LTR10A/F, LTR8B is not labelled. Why? I see JUNB is among the motifs enriched at LTR8B in Fig S2A, maybe this would be better to reference here? If indeed it is also enriched in 2C/S2B it would be useful to add a label in Fig 2C/S2B, if not it would be good to explain what made you look for JUN motifs at LTR8B elements: are there just very few JUN motifs/JUN-activated LTR8B elements, as indicated later in line 286 by the low number of changed target genes? I think this is important as LTR8B is analyzed in more detail in the remainder of the manuscript.

- line 251f: the large fraction of genes nearby ERV-derived enhancers expressed in SynT vs. other placental cell types is intriguing. The authors give a nice example of a cluster of descendants of an ancestral ERV-driven PSG gene as a partial explanation for this enrichment. There seem to be several yellow lines (highly expressed genes) on the plot in Fig 3C other than PSG genes: are the other genes also clusters of related genes driven by a subfamily of ERVs?

- line 272f the conclusion of increased cell migration genes could be strengthened by not just providing GO terms in FigS4 but e.g. scatter or MA plots showing all up and downregulated genes and biogwig tracks of loci of interest such as cell migration genes; this would make it easier to believe gene expression changes. Can the authors speculate why JNK inhibition has opposite effects in trophoblast and cancer cells, while they mention later in the discussion that there are similarities between cancer and trophoblast?

- line 275-83: Is increased JUN expression upon JNK inhibition a known effect of JNK inhibition or unexpected? If known, adding a reference here would strengthen this result. If unexpected: explain what would be expected? I like that possible mechanisms are discussed, but they are difficult to understand (maybe this is just because of the complexity of JNK activity that the authors mention in the acknowledgements? Or could it maybe be explained better here).

- line 334: 'out of 12 genes' > it would be helpful to refer the reader to a list of the 12 genes identified and their mTSC expression, as Figure 4E just shows only 1 of them. may have missed this data, but if it is missing, please add it.

- Table S1 has some of the 29 identified repeat families highlighted in bold blue: what are these? I could not find a legend in the file. In legend for Figure S1 blue means shared significant between 2 cell types. Is this also the case here? I thought all 29 were shared (see line 118f main text)? In the main text line 140f 'we decided to focus on 18 placenta-specific candidate regulatory repeat families, all of which were ERV-associated LTRs (Table S1)'. It is not clear if the bold ones are the ERV-associated ones or were chosen for another reason? Please clarify.

- Figures 1C and S1A: the authors could consider adding the number of bed regions for each heatmap (how many repeats are in each family?).

- legend S3A: typo disrance > distance
- table S2: ctrl2,4,1,3 labels: which cell lines/samples is this expression data for? Based on Fig S3A mentioned together with this table in lines 228-232 these samples are likely primary cytotrophoblast, hTSC, SynT and EVT? line 232 should probably say "were lowly expressed in human placental cell types (cytotrophoblast, hTSC, SynT and EVT)" (Table S2). similarly, line 229 could say "whose expression in <human placental cell types> is driven by a MER39 element"
- I may have missed this, but the supplementary tables seem to lack legends: I found it difficult to understand what e.g. Table S2 is from lines 201 in the text
- line 312 'as expected....' it may be helpful for the reader to refer back to Fig4B here or say something like "LTR2B, the human ERV subfamily with the least orthologous sequences in other primates"
- Fig 4 legend: typo putateive > putative
- Supplementary Figure 5 is labelled 4.
- line 432: in striking a decrease > in a striking decrease

Reviewer #2:

Remarks to the Author:

In this manuscript from Branco and co-workers the authors explore a role for ERVs as lineage-specific placental enhancers. This is an interesting, comprehensive, and carefully conducted study that is well-planned and executed. The authors use chromatin marks to identify ERVs with gene regulatory capacity and are located in close proximity to trophoblast-expressed genes and then use CRISPR excision to proof that the identified ERVs truly act as gene enhancers. This represents a significant advance to the field. Overall, these are important and novel findings that connects regulatory ERVs to pregnancy complications and provides a basis for future studies and therapeutic targets. Most of the results appear very solid. However, there are a few concerns that should be addressed in a revised version.

Major points:

1. The authors claim that they analysed primary human cytotrophoblast and human trophoblast stem cells (hTSCs) but in most figures only hTSCs are shown (e.g. Figure 1B). I would suggest to include the chromatin mark data for cytotrophoblast cells shown in Supplementary table 1 in the figures or clarify why the data is not shown in the figures.
2. It was not clear to me if the macaque RNA-seq reads were only aligned to hg38 or also to the macaque reference genome? Higher gene expression of genes close to human-specific ERVs could otherwise be due biases, for example in the reference genome builds or mapping (if aligning only to hg38)? Could the authors clarify this point?
3. CRISPR excision: Please include PCR gels to show that excision worked (Figure 5).
4. qPCR: Please clarify if the replicates shown in the figures are technical or biological replicates? Also please clarify why some genes seem to have more replicates than others (e.g. Figure 5: ADAM9 (P1 = 4 data points) versus TWIST1 (8 data points))? Were expression levels confirmed with additional primer pairs?

Minor points:

- Figure 1E: It is very difficult to read the labels.
- Supplementary figure 1C: Please add scale bars. What is the blue staining (DAPI)?
- Figure 2D: It is not possible to read the sequence and sequence labels.
- p. 10, line 202: Why was LTR8B selected? Maybe highlight in previous graph?
- Supplementary figure 5 is mislabeled as Supplementary figure 4
- Supplementary figure 5A: It could be beneficial to add the p values even though the difference is not statistically significant.
- Supplementary figure 6C: Please add scale bars.

Reviewer #3:

Remarks to the Author:

In this manuscript, Frost et al. describe enhancers and promoters derived from endogenous retroviruses with significant contributions to human trophoblast gene regulation. The study shows that TEs, and particularly primate-specific ERV/LTRs, are an important source of gene regulatory elements, and cluster near genes with preferential trophoblast expression in a species-specific manner. Overall, this is an interesting and high quality survey of ERV activity in trophoblast cells and their contribution to placental development and evolution. The manuscript is well written and the results are well presented.

However, the novelty of these findings is somewhat limited when compared to other studies (e.g. Dunn-Fletcher et al, 2018 Plos Bio; Chuong 2018 Plos Bio; Deniz et al 2020 Nat Comm; Sun et al 2021 MBE; Fueyo et al 2022 Nat Rev Mol Cell Biol). Many of these previous studies also performed functional validation to show that ERVs drive the expression of nearby genes in a species-specific and cell-type-specific manner, also focusing on placental ERV-derived enhancers. Similarly, the enrichment of placenta-associated transcription factor motifs within ERV families (e.g. STAT/SRF for MER41B elements) has been previously reported.

The main novelty here is a re-analysis of existing data, including a comprehensive overview of many ERV families with gene regulatory hallmarks in human trophoblast (e.g. MER41B, LTR10A/F, LTR8B, LTR3A, MER61E, etc), as well as specific CRISPR-validated examples of ERVs that regulate essential placental developmental genes, such as the LTR10A element within the ENG gene. The ENG case example is potentially significant due to previously established links with preeclampsia. In terms of activating mechanism, the authors tried to investigate ERVs with JUN motifs (e.g. LTR10A/F and LTR8B elements) by treating cells with SP600125, a JNK inhibitor. However, the results were somewhat contradictory, since JNK inhibition led to an increase in JUN expression (rather than a decrease) and nearby JUN target genes were likewise upregulated. No attempt was made to validate this result with other JNK/JUN inhibitors or activators, or even a CRISPR knockdown of JUN itself, to observe how this affects nearby ERV-regulated or JUN-dependent genes.

As such, I have a few comments on the methodology that the authors may wish to address to improve the overall robustness of the manuscript, as well as minor comments to improve clarity or add support.

Major Comments:

1) The authors use distance as the main metric by which to associate target genes with candidate ERV enhancers (e.g. 50 kb for hTSC-active ERVs, or 100 kb for hTSC-active ERVs shared between human and macaque or mouse). In Figure 3A, 3B, 3F, Fig 4C, 4D, these analyses lack sufficient detail in the figure legends/methods sections. How many genes were in each class, were they filtered in any way (and were annotated genes with zero counts included?), etc, should all be added. Aside from including this missing information, these RNA-seq analyses should be made more robust. For example, numerous studies suggest that enhancers (ERV-derived or otherwise) can have long-range effects on gene expression (e.g. Kleinjan et al 2005 Am J Hum Gen; Fuentes et al 2018 Elife; Raviram et al 2018 Genome Bio). The authors may want to incorporate available Hi-C datasets, or a combination of ChIP-seq/RNA-seq/Hi-C (as suggested by the Activity-By-Contact model developed by Fulco et al 2019 Nat Gen) to identify all possible genes regulated by these placental ERV enhancers.

2) The mechanism by which ERVs are activated as enhancers in human trophoblast cells is poorly investigated. The only experiment that attempts to address this is the SP600125 treatment in hTSCs and subsequent RNA-seq. However, the results are inconclusive, since the authors observe that genes lying within 5 kb of a JUN binding site appear to be upregulated after treatment with SP600125. The analyses presented in Figs 3E and 3F are not very convincing and also not sufficiently described (how many genes in each set, etc?). It would also be more convincing if the families themselves showed a difference in regulation at the family transcript level, as measured by a program like TETranscripts or SalmonTE. The authors mention that JNK1 and JNK2 are observed to sometimes have opposing roles and this may affect JNK signaling. In that case, the authors may want to consider using a different JUN or AP1 inhibitor. If no further experiments are performed to clarify this result, the authors should consider removing this section or moving it to supplementary due to the ambiguous results.

Minor Comments:

1) Line 160: "It is possible that a different set of poised ERV enhancers become active upon differentiation into SynT" Was any attempt made to find the SynT-specific set of enhancers? More generally, do any ERV-derived enhancers that are seemingly active in hTSCs become inactive in differentiated cells? Or does differentiation always lead to an increase in ERV enhancer activity? (e.g. gain of H3K27ac marks on formerly poised enhancer regions)

2) Line 177: "Namely, multiple families bore motifs for key factors involved in the maintenance of the stem cell state, such as ELF5, GATA3..." Did unmarked elements also harbor these transcription factor motifs? Or were the motifs only found in H3K27ac-marked ERVs? Please clarify.

3) Line 193: "We also found instances of enriched transcription factor binding to families that seemingly do not bear the corresponding motif" Is there any evidence of these TFs motifs by local alignment, or by relaxing the p-value threshold of the motif finding program (e.g. FIMO)? Or are there other motifs present on these ERV families?

4) Line 210: "In the case of LTR8B, only active elements contained one AP-1 motif" Were there any noticeable differences in JUN/JUND binding when comparing LTR8B elements (with one AP1 motif) to LTR10A/F elements (with three AP-1 motifs)?

5) Line 283: "LTR10A (but not LTR10F) target genes were upregulated upon SP600125 treatment" Are there any motif or sequence-specific differences between LTR10F and LTR10A elements that might explain why LTR10F elements do not respond to SP600125 treatment? Similarly, are LTR10F elements repressed in trophoblast cells? (e.g. any evidence of LTR10F-specific silencing via H3K9me3 ChIP-seq or ZNFs?)

6) Line 328: "one human-specific ERV was associated with expression of a Wnt signalling receptor (FZD5)" Since this is a single ERV, perhaps include in parentheses the family that this ERV belongs to (eg. MER39, LTR8B, LTR10A, or a different family?)

7) Line 380: "CSF1R expression of the placenta-specific variant being reduced by around 2-fold" The LTR10A deletion is shown to significantly affect CSF1R gene expression in hTSCs. Since JUN/JUND motifs are thought to activate LTR10A enhancers, and CSF1R is an LTR10A-regulated gene, the authors should clarify whether SP600125 treatment in hTSCs also affected CSF1R gene expression (as expected).

8) Line 507: "the combination of an arguably permissive chromatin conformation and shared signalling pathways may make the co-option of ERVs for both placental development/function and cancer a frequent occurrence" Consider expanding this section, e.g. which signalling pathways are usually activated in both cancers and placental development? Are the same mechanisms thought to activate these ERV families in both placental development and cancer progression?

9) Line 809: "Motifs enriched at active TE families were identified using the AME tool of the MEME suite" Was any differential enrichment motif analysis performed? E.g. comparing active MER11B elements to inactive (unmarked) MER11B elements?

10) In general, the methods section could use more detail to ensure reproducibility. This includes e.g. specifying which (non-default) flags/parameters were used for each program; why broad peaks were used rather than narrow peaks for MACS peak calling; the p-value thresholds using with FIMO and the MEME suite; and the type of normalization used with DESeq2 to perform differential expression analysis. All programs used should also be referenced.

Author Rebuttal to Initial comments

We sincerely appreciate the reviewers' thorough assessment of our work, and thank them for taking the time to highlight its strengths and make useful suggestions. We address their comments and suggestions point by point below.

Reviewer #1:

- the authors do not discuss if there is anything unique about the repeat families co-opted in placenta vs hESCs, which may be relevant given that placental morphology is generally considered to have evolved more rapidly than embryos, and as the authors point out that TEs may fuel fast placental evolution across primates (in Fig4A and lines 299.) While an analysis of orthologous regions (evolutionary conservation) is done in Fig4B for primate species, it would be useful to assess ages of TEs coopted for hTSCs vs hESCs. Are the cooped TEs in TSC/ESC younger/older? less conserved/similarly conserved? more ERVs than other TE families or similar between

ESC/TSC? for example: what distinguishes the coopted LTR7up hESC and LTR7C hTSC coopted TEs? Age? Sequence?

This is an interesting point, but we do not believe that a comparison with hESCs alone would be sufficient to address this question. hESCs represent a transient state in early development, whereas most embryonic development occurs after the blastocyst stage, via multiple lineages that have their own landscape of active TEs (<https://www.biorxiv.org/content/10.1101/2021.08.18.456764v1>). The comparison with the much simpler placental development is therefore made complicated, and we believe a thorough analysis is beyond the scope of this paper. Additionally, the crux of the question lies in what proportion of these elements is truly functional and affects phenotypes.

For the reviewer's interest, we performed a TE orthology analysis (which provides a more accurate picture of TE age than analyses based on mutation rates) on TE families with regulatory potential in hESCs and found no obvious difference in TE ages when compared to hTSCs (compare figure below with Figure 4B of the manuscript). We also see no major difference in the total number of enhancer-like TEs between the two cell types.

As for the LTR7C/LTR7up subfamilies, they have indeed sequence differences, which have been explored in great detail in PMID:35179489. LTR7up activity in hESCs is driven by a SOX2/3 binding motif that is absent in LTR7C. LTR7C also contains a unique combination of motifs (including for TEAD4 and AP2-gamma), which may drive trophoblast-specific expression, although this would have to be tested experimentally. We have added the following sentence to our manuscript (lines 124-126): "Each of the LTR7 families displays a unique combination of transcription factor binding motifs (Carter et al., 2022), which may underlie their cell type-specific expression".

- Line 131: 'more specifically, it is the LTR7up subfamily that is active in hESCs' - Figure S1A does only show LTR7 elements: was it challenging to display LTR7up subfamily elements or why is this statement included in the main text here but not represented in the data included here (LTR7up must be contained in the LTR7 peaks?) If possible, the authors could add a subset heatmap with the LTR7up repeat elements?

We had not performed this analysis ourselves and were simply referring to results from the paper cited (PMID:35179489), which is why we were careful to place this statement after the reference to the Figure. Having now performed our own analysis we found that, in fact, H3K27ac signal can be seen not just in LTR7up elements, but in LTR7d elements as well:

A more careful examination of the cited paper actually shows consistency with this. For whilst the transcriptional signal and H3K4me3 enrichment in hESCs is specific to the LTR7up subfamily, all LTR7 subfamilies display some level of H3K27ac enrichment (Figure 3B of PMID:35179489). As the intricacies of the LTR7 subfamilies are not the focus of our work, and have been extensively explored in the cited paper, we have opted for removing the sentence in question.

- line 172f: 'focused on a selection of transcription factors that are expressed in trophoblast' How were these transcription factors and their expression identified (expression cut-off or just picked examples of known placentally expressed TFs)?

We have added the following text to the methods text to clarify this point (lines 832-835):

"Relevant motifs were selected based on the expression of the respective transcription factors in hTSC ($>1 \log_2(\text{FPKM})$), followed by clustering to find redundant motifs and further selection based on literature searches". We also added reference to the Methods section in the relevant point in the Results text. Our Github repository includes the script that was used for this selection ahead of literature searches.

- Line 206-212: 'active elements' I suspect this means marked by certain marks (H3K4me1/H3K27ac) in hTSCs? Please clarify what active means (e.g. marked by H3K4me1/H3K27ac in hTSCs)

We mean elements marked by H3K27ac. We clarified this in lines 202-203: "Both LTR10A and LTR10F active elements (i.e., those marked by H3K27ac)...".

- line 283: judging from Fig 2C both LTR10A and LTR10F are enriched for JUN binding so I would expect target genes of both subfamilies to contribute to observed gene expression changes upon JNK inhibition. Can the authors speculate why only target genes of one change?

It may be that LTR10F target genes are also affected, but there was insufficient statistical power to be certain, given the few genes associated with this family. Nonetheless, the more pronounced effect at LTR10A-associated genes is in line with the higher enrichment of JUN/JUND at these elements when compared to LTR10F. We have added text to the respective results section (lines 284-287) making both of these points.

- line 201/202: the authors reference Fig 2C and S2B in which, unlike LTR10A/F, LTR8B is not labelled. Why? I see JUNB is among the motifs enriched at LTR8B in Fig S2A, maybe this would be better to reference here? If indeed it is also enriched in 2C/S2B it would be useful to add a label in Fig 2C/S2B, if not it would be good to explain what made you look for JUN motifs at LTR8B elements: are there just very few JUN motifs/JUN-activated LTR8B elements, as indicated later in line 286 by the low number of changed target genes? I think this is important as LTR8B is analyzed in more detail in the remainder of the manuscript.

We appreciate the suggestion. We have now labelled the LTR8B family in Figures 2C and S2B.

- line 251f: the large fraction of genes nearby ERV-derived enhancers expressed in SynT vs. other placental cell types is intriguing. The authors give a nice example of a cluster of descendants of an ancestral ERV-driven PSG gene as a partial explanation for this enrichment. There seem to be several yellow lines (highly expressed genes) on the plot in Fig 3C other than PSG genes: are the other genes also clusters of related genes driven by a subfamily of ERVs?

This is an interesting question, but the PSG cluster looks to be a special (and very interesting) case. All other genes that are highly expressed in SynT are not in clusters and are associated with a few different TE families.

- line 272f the conclusion of increased cell migration genes could be strengthened by not just providing GO terms in FigS4 but e.g. scatter or MA plots showing all up and downregulated genes and biogwig tracks of loci of interest such as cell migration genes; this would make it easier to believe gene expression changes. Can the authors speculate why JNK inhibition has opposite effects in trophoblast and cancer cells, while they mention later in the discussion that there are similarities between cancer and trophoblast?

We have added a volcano plot (Supp. Fig. 4C) and two genome browser snapshots of genes affected by SP600125 treatment (Supp. Fig. 4D). Regarding the effects of JNK inhibition, we now show that it leads to increased levels of phosphorylated c-Jun/JUN in the nucleus (Supp. Fig. 4A), which is known to occur upon JNK2 inhibition. This is in line with high expression of JNK2 in hTSCs (Supp. Fig. 4B). Differences in the expression of JNK1 and JNK2 (which have opposite effects) could explain some cell-specific effects. However, the aim of our experiments was to test whether ERV-associated genes were regulated by AP-1, and a detailed discussion of the precise upstream mechanisms are outside of the remit of our paper.

- line 275-83: Is increased JUN expression upon JNK inhibition a known effect of JNK inhibition or unexpected? If known, adding a reference here would strengthen this result. If unexpected: explain what would be expected? I like that possible mechanisms are discussed, but they are difficult to understand (maybe this is just because of the complexity of JNK activity that the authors mention in the acknowledgements? Or could it maybe be explained better here).

As described in reply to the above point, we now have a much clearer explanation as to why we observed upregulation of AP-1 target genes upon JNK inhibition, and have revised this section of the manuscript accordingly (lines 267-279).

- line 334: 'out of 12 genes' > it would be helpful to refer the reader to a list of the 12 genes identified and their mTSC expression, as Figure 4E just shows only 1 of them. may have missed this data, but if it is missing, please add it.

We have included a new Supplementary Table (S3) that contains detailed information on all 12 genes and associated TEs.

- Table S1 has some of the 29 identified repeat families highlighted in bold blue: what are these? I could not find a legend in the file. In legend for Figure S1 blue means shared significant between 2 cell types. Is this also the case here? I thought all 29 were shared (see line 118f main text)? In the main text line 140f 'we decided to focus on 18

placenta-specific candidate regulatory repeat families, all of which were ERV-associated LTRs (Table S1)'. It is not clear if the bold ones are the ERV-associated ones or were chosen for another reason? Please clarify.

We apologise for the confusion. The highlighted families are the 18 placenta-specific ERV families that we focused on for the rest of the paper. We have clarified this in the legend to the table.

- Figures 1C and S1A: the authors could consider adding the number of bed regions for each heatmap (how many repeats are in each family?).

Useful suggestion, thank you. We have added the number of TEs next to each heatmap.

- table S2: ctrl2,4,1,3 labels: which cell lines/samples is this expression data for? Based on Fig S3A mentioned together with this table in lines 228-232 these samples are likely primary cytotrophoblast, hTSC, SynT and EVT? line 232 should probably say "were lowly expressed in human placental cell types (cytotrophoblast, hTSC, SynT and EVT)" (Table S2). similarly, line 229 could say "whose expression in <human placental cell types> is driven by a MER39 element"

These are indeed data from primary cytotrophoblast. The nomenclature was to do with the fact that the dataset also contains cells treated with vitamin D. We have now changed this. We have also made changes to the text as suggested (lines 228 and 231).

- I may have missed this, but the supplementary tables seem to lack legends: I found it difficult to understand what e.g. Table S2 is from lines 201 in the text

We apologise for this omission. We have now added table legends to the top of each excel sheet.

- line 312 'as expected....' it may be helpful for the reader to refer back to Fig4B here or say something like "LTR2B, the human ERV subfamily with the least orthologous sequences in other primates'

We appreciate the suggestion. We have changed the sentence to: "The majority of non-orthologous elements were from the LTR2B family (as expected from its near absence in macaque; Figure 4B)...".

- legend S3A: typo disrance > distance
- Fig 4 legend: typo putateive > putative
- Supplementary Figure 5 is labelled 4.
- line 432: in striking a decrease > in a striking decrease

These typos have been corrected.

Reviewer #2:

1. The authors claim that they analysed primary human cytotrophoblast and human trophoblast stem cells (hTSCs) but in most figures only hTSCs are shown (e.g. Figure 1B). I would suggest to include the chromatin mark data for

cytotrophoblast cells shown in Supplementary table 1 in the figures or clarify why the data is not shown in the figures.

We appreciate the suggestion. We now include the cytotrophoblast H3K27ac enrichment data in Supplementary Figure 1A. As these data were generated with a different technique (ChIP-seq), we also show results from ChIPseq on hTSCs.

2. It was not clear to me if the macaque RNA-seq reads were only aligned to hg38 or also to the macaque reference genome? Higher gene expression of genes close to human-specific ERVs could otherwise be due biases, for example in the reference genome builds or mapping (if aligning only to hg38)? Could the authors clarify this point?

The macaque RNA-seq data was only aligned to the macaque reference genome. We have updated our Methods section to clarify this: "Mapping was done with either Bowtie2 (for CUT&Tag, CUT&RUN and ChIP-seq; default settings) (Langmead and Salzberg, 2012) or Hisat2 (for RNA-seq; with --no-softclip) (Kim et al., 2019) to the reference genome of the species of origin: hg38, mm10 or rheMac10."

3. CRISPR excision: Please include PCR gels to show that excision worked (Figure 5).

We appreciate the reviewer's suggestion, but we had already included genotyping qPCR data in our original submission (Supplementary Table S4), which gives quantitative information about the proportion of excised alleles in each cell pool used. We show below examples of gel images to reassure the reviewer that we obtained bands of the expected sizes. But we have refrained from including endpoint PCR data in the manuscript, which is not quantitative, and may mislead readers.

4. qPCR: Please clarify if the replicates shown in the figures are technical or biological replicates? Also please clarify why some genes seem to have more replicates than others (e.g. Figure 5: ADAM9 (P1 = 4 data points) versus TWIST1 (8 data points))? Were expression levels confirmed with additional primer pairs?

The RT-qPCR replicates are a combination of independent lentiviral infections (with the same or different sgRNA sets, as shown in Supplementary Table S4) and collection of RNA at different passage numbers (from the same infection). We now include in the legends to Figures 5 and 6 the number of sgRNA sets and independent infections used for each plot. We have also updated the methods section to better clarify this: "The number of independent RT-qPCR replicates used is visible on the respective plots – these were derived using different sgRNA sets or independent infections (as detailed in the figure legends and Supplementary Table S4), and/or from RNA collections at different passage numbers."

The number of data points simply reflect differences in the number of infections and qPCRs performed, which were not homogeneous across targets. All primer pairs used and associated data are included in the manuscript.

Minor points:

- Figure 1E: It is very difficult to read the labels.

We have made these labels larger.

- Supplementary figure 1C: Please add scale bars. What is the blue staining (DAPI)?

We have added scale bars and clarified (in the images and legend) that the blue staining is indeed DAPI.

- Figure 2D: It is not possible to read the sequence and sequence labels.

We have made the font larger for sequences and labels.

- p. 10, line 202: Why was LTR8B selected? Maybe highlight in previous graph?

Good point. We have highlighted LTR8B in Figure 2C.

- Supplementary figure 5 is mislabeled as Supplementary figure 4

Thank you for pointing it out. We have corrected this.

- Supplementary figure 5A: It could be beneficial to add the p values even though the difference is not statistically significant.

We have added these p values.

- Supplementary figure 6C: Please add scale bars.

We have added scale bars.

Reviewer #3:

1) The authors use distance as the main metric by which to associate target genes with candidate ERV enhancers (e.g 50 kb for hTSC-active ERVs, or 100 kb for hTSC-active ERVs shared between human and macaque or mouse). In Figure 3A, 3B, 3F, Fig 4C, 4D, these analyses lack sufficient detail in the figure legends/methods sections. How many genes were in each class, were they filtered in any way (and were annotated genes with zero counts included?), etc, should all be added. Aside from including this missing information, these RNA-seq analyses should be made more robust. For example, numerous studies suggest that enhancers (ERV-derived or otherwise) can have long-range effects on gene expression (e.g. Kleinjan et al 2005 Am J Hum Gen; Fuentes et al 2018 Elife; Raviram et al 2018 Genome Bio). The authors may want to incorporate available Hi-C datasets, or a combination of ChIP-seq/RNA-seq/Hi-C (as suggested by the Activity-By-Contact model developed by Fulco et al 2019 Nat Gen) to identify all possible genes regulated by these placental ERV enhancers.

We apologise for the lack of detail. We have added the number of genes in each of the indicated figure panels, and have updated the methods section to clarify our filtering: “For all analyses, log2 fold differences were calculated using only genes that passed a minimal expression threshold (variable depending on dataset and normalisation strategy) in at least one of the two samples compared”. For those interested in further details, we have deposited all our scripts on GitHub (https://github.com/MBrancoLab/Frost_2022_hTroph).

The inclusion of Hi-C data is a very pertinent suggestion. However, there is a lack of suitable datasets in human trophoblast. We are only aware of one relevant promoter capture Hi-C dataset from trophoblast-like cells (PMID: 31501517). These are cells derived from ‘primed’ hESCs by treatment with BMP4, which display some characteristics of trophoblast (PMID: 12426580). However, these trophoblast-like cells substantially differ from primary trophoblast, and their expression profile more closely resembles that of amnion (PMID: 33831365). Having had already considered this ahead of the initial submission, we did not feel that it would be an appropriate dataset to use. Whilst we are aware that our stringent proximity criteria inevitably excluded some relevant genes, it also minimised noise, allowing us to pick the most promising candidates to validate by CRISPR.

2) The mechanism by which ERVs are activated as enhancers in human trophoblast cells is poorly investigated. The only experiment that attempts to address this is the SP600125 treatment in hTSCs and subsequent RNAseq. However, the results are inconclusive, since the authors observe that genes lying within 5 kb of a JUN binding site appear to be upregulated after treatment with SP600125. The analyses presented in Figs 3E and 3F are not very convincing and also not sufficiently described (how many genes in each set, etc?). It would also be more convincing if the families themselves showed a difference in regulation at the family transcript level, as measured by a program like TETtranscripts or SalmonTE. The authors mention that JNK1 and JNK2 are observed to sometimes have opposing roles and this may affect JNK signaling. In that case, the authors may want to consider using a different JUN or AP1 inhibitor. If no further experiments are performed to clarify this result, the authors should consider removing this section or moving it to supplementary due to the ambiguous results.

We agree with the reviewer that the experiments with the JNK inhibitor had some unanswered questions. We have performed additional experiments and can now demonstrate that SP600125 actually leads to increased levels of phosphorylated c-Jun/JUN in the nucleus of hTSCs (Supp. Fig. 2A). We obtained the same result with a second AP-1 inhibitor (Supp. Fig. 2A). We speculate that this is due to inhibition of JNK2, which (in contrast with JNK1 deficiency) is known to lead to increased phosphorylation and stability of JUN, and which is highly expressed in hTSCs (Supp. Fig. 2B). We have revised the text accordingly (lines 267-279). As suggested, we have also added information on the number of genes in Figures 3E/F (in 3F we added the individual data points to the plot).

Finally, we took up the reviewer’s interesting suggestion of assessing ERV transcription and found that LTR10A driven transcription of internal ERV regions can indeed be seen upon SP600125 treatment (Supp. Fig. 4G), supporting the claim that the regulatory activity of LTR10A is partly controlled by AP-1. We found no LTR8B elements with a proviral arrangement to test this on.

Minor Comments:

1) Line 160: “It is possible that a different set of poised ERV enhancers become active upon differentiation into SynT”
Was any attempt made to find the SynT-specific set of enhancers? More generally, do any ERV-derived enhancers that are seemingly active in hTSCs become inactive in differentiated cells? Or does differentiation always lead to an increase in ERV enhancer activity? (e.g. gain of H3K27ac marks on formerly poised enhancer regions)

There were indeed several attempts to identify SynT-specific enhancers, as explained in the results section (lines 153-156): “It is possible that a different set of poised ERV enhancers become active upon differentiation into syncytiotrophoblast (SynT; Supplementary Figure 1C), but the multinucleated nature of these cells seemingly interfered with our CUT&Tag attempts”.

We did initially ask whether hTSC-active ERV-derived enhancers became inactive in EVT. For the reviewer’s consideration, an example of this analysis is displayed below, where we show the state of each MER21-derived enhancer in both hTSC and EVT. However, we were conscious that the CUT&Tag data for hTSC is of substantially better quality than that for EVT, thus risking that some of the silenced enhancers in EVT could be false negatives. This is why we were very careful to focus our analysis and conclusions only on EVT-active enhancers.

- 2) Line 177: “Namely, multiple families bore motifs for key factors involved in the maintenance of the stem cell state, such as ELF5, GATA3...” Did unmarked elements also harbor these transcription factor motifs? Or were the motifs only found in H3K27ac-marked ERVs? Please clarify.

To answer this question we have performed transcription motif analysis on H3K27ac-negative elements from the same families (Supplementary Figure 2B), and added the following text to the results section (lines 174-179): “Most of these motifs were also present in elements negative for H3K27ac (Supplementary Figure 2B), suggesting that there are additional genetic or epigenetic determinants of their activity, similar to what we had observed in mouse TSCs (Todd et al., 2019). There were nonetheless some notable exceptions where motifs were only found in active elements, such as GATA3 in LTR2B elements and SRF in MER61D elements (Supplementary Figure 2B)”.

- 3) Line 193: “We also found instances of enriched transcription factor binding to families that seemingly do not bear the corresponding motif” Is there any evidence of these TFs motifs by local alignment, or by relaxing the pvalue threshold of the motif finding program (e.g. FIMO)? Or are there other motifs present on these ERV families?

The most obvious example in this respect is the LTR3A family, which is enriched for GATA3 and TFAP2C binding (Figure 2C), but seemingly does not bear the corresponding motifs (Figure 2A). Having now lowered the FIMO p-value threshold from $1e-4$ to 0.01 we still do not find any GATA3 or TFAP2C motifs at active LTR3A elements. These elements do however harbour TEAD4 motifs (Figure 2A) and are enriched for TEAD4 binding (Figure 2B), which may recruit other transcription factors. For example, mouse TEAD4 and TFAP2C have been shown to form a complex (PMID: 32197068). We have added this reference to the relevant results section (lines 192-196): “This could reflect limitations of motif-finding approaches and/or suggest that interactions between different transcription factors enable recruitment of large regulatory complexes based on a small subset of motifs (e.g., TEAD4, which binds to motifs in LTR3A elements, may recruit TFAP2C (Chi et al., 2020)).

4) Line 210: “In the case of LTR8B, only active elements contained one AP-1 motif” Were there any noticeable differences in JUN/JUND binding when comparing LTR8B elements (with one AP1 motif) to LTR10A/F elements (with three AP-1 motifs)?

Yes. The reviewer should be able to appreciate those differences when comparing the JUN/JUND enrichments in Figure 2E (notice the different y-axis scales). Why this difference in enrichment exists is unclear. The AP-1 motifs are slightly different in sequence and arrangement, but there are also other motif differences between the two families (Supp. Fig. 2A). We describe a few of these possibilities in the manuscript (lines 207-210).

5) Line 283: “LTR10A (but not LTR10F) target genes were upregulated upon SP600125 treatment” Are there any motif or sequence-specific differences between LTR10F and LTR10A elements that might explain why LTR10F elements do not respond to SP600125 treatment? Similarly, are LTR10F elements repressed in trophoblast cells? (e.g. any evidence of LTR10F-specific silencing via H3K9me3 ChIP-seq or ZNFs?)

It may be that LTR10F target genes are also affected, but there was insufficient statistical power to be certain, given the few genes associated with this family. Nonetheless, the more pronounced effect at LTR10A-associated genes is in line with the higher enrichment of JUN/JUND at these elements when compared to LTR10F. We have added text to the respective results section (lines 284-287) making both of these points.

The role of repressive marks in balancing ERV regulatory activity is an interesting topic. Most ERV families studied here have a substantial proportion of elements covered by H3K9me3 (examples in Fig. 1E). However, in the specific case of LTR10A/LTR10F, this does not explain why LTR10A is the more active family, as it is also more enriched for H3K9me3, as shown below (and a higher proportion of elements are marked by H3K9me3).

6) Line 328: “one human-specific ERV was associated with expression of a Wnt signalling receptor (FZD5)” Since this is a single ERV, perhaps include in parentheses the family that this ERV belongs to (eg. MER39, LTR8B, LTR10A, or a different family?)

We have added this information (it is a MER41B element).

7) Line 380: “CSF1R expression of the placenta-specific variant being reduced by around 2-fold” The LTR10A deletion is shown to significantly affect CSF1R gene expression in hTSCs. Since JUN/JUND motifs are thought to activate LTR10A enhancers, and CSF1R is an LTR10A-regulated gene, the authors should clarify whether SP600125 treatment in hTSCs also affected CSF1R gene expression (as expected).

Thank you for pointing this out. *CSF1R* and *ENG* (also an LTR10A target) are indeed both upregulated upon SP600125 (Supp. Fig. 4F). We have added the corresponding text to lines 376-378 and 437-439.

- 8) Line 507: “the combination of an arguably permissive chromatin conformation and shared signalling pathways may make the co-option of ERVs for both placental development/function and cancer a frequent occurrence” Consider expanding this section, e.g. which signalling pathways are usually activated in both cancers and placental development? Are the same mechanisms thought to activate these ERV families in both placental development and cancer progression?

We have expanded this section slightly (lines 501-507): “The combination of an arguably permissive chromatin conformation and shared signalling pathways (Costanzo et al., 2018) may make the co-option of ERVs for both placental development/function and cancer a frequent occurrence. For example, the Hippo signalling pathway, acting through YAP and TEAD4, plays key roles in both tumourigenesis and placental development (Meinhardt et al., 2020). The activation of LTR10A elements in both contexts is also seemingly driven by a shared signalling pathway, in this case MAPK/AP-1 (Ivancevic et al., 2021).”

- 9) Line 809: “Motifs enriched at active TE families were identified using the AME tool of the MEME suite” Was any differential enrichment motif analysis performed? E.g. comparing active MER11B elements to inactive (unmarked) MER11B elements?

As a result of our reply to the reviewer’s comment above (minor comment number 2), Supplementary Figure 2 can now be used to compare motif enrichment between active and inactive elements, and we have highlighted a few striking differences therein.

- 10) In general, the methods section could use more detail to ensure reproducibility. This includes e.g. specifying which (non-default) flags/parameters were used for each program; why broad peaks were used rather than narrow peaks for MACS peak calling; the p-value thresholds using with FIMO and the MEME suite; and the type of normalization used with DESeq2 to perform differential expression analysis. All programs used should also be referenced.

We apologise for these omissions. We have added details on the parameters used for Trim_galore, Bowtie2, Hisat2, deepTools, DESeq2, AME, FIMO and Stringtie. For those interested in further details, we have deposited all our scripts on GitHub (https://github.com/MBrancoLab/Frost_2022_hTroph). We have also added the relevant references.

We used the broad peak option of MACS, as this is what is generally recommended for histone marks (<https://github.com/macs3-project/MACS>; note that only histone marks were analysed with MACS), and thus we do not feel a justification needs to be added to the manuscript. Histones yield broader peaks than transcription factors, and we do find that the --broad option makes for more adequate peak definition.

Decision Letter, first revision:

Message: Our ref: NSMB-A46359A

15th Dec 2022

Dear Dr. Branco,

Thank you for submitting your revised manuscript "Regulation of human trophoblast gene expression by endogenous retroviruses" (NSMB-A46359A). It has now been seen by the original referees and their comments are below. The reviewers find that the paper has improved in revision, and therefore we'll be happy in principle to publish it in Nature Structural & Molecular Biology, pending minor revisions to comply with our editorial and formatting guidelines.

Sincerely,

Carolina

Carolina Perdigoto, PhD
Chief Editor
Nature Structural & Molecular Biology
orcid.org/0000-0002-5783-7106

Reviewer #1 (Remarks to the Author):

The authors have adequately addressed my questions.

Reviewer #2 (Remarks to the Author):

The authors have addressed all comments. The manuscript is ready to be published.

Reviewer #3 (Remarks to the Author):

The authors have addressed both my major and minor concerns. I have no further comments.

Final Decision Letter:

Message 2nd Mar 2023

:

Dear Dr. Branco,

We are now happy to accept your revised paper "Regulation of human trophoblast gene expression by endogenous retroviruses" for publication as a Article in Nature Structural & Molecular Biology.

Your paper will be published online soon after we receive proof corrections and will appear in print in the next available issue. You can find out your date of online publication by contacting the production team shortly after sending your proof corrections. Content is published online weekly on Mondays and Thursdays, and the embargo is set at 16:00 London time (GMT)/11:00 am US Eastern time (EST) on the day of publication. Now is the time to inform your Public Relations or Press Office about your paper, as they might be interested in promoting its publication. This will allow them time to prepare an accurate and satisfactory press release. Include your manuscript tracking number (NSMB-A46359B) and our journal name, which they will need when they contact our press office.

About one week before your paper is published online, we shall be distributing a press release to news organizations worldwide, which may very well include details of your work. We are happy for your institution or funding agency to prepare its own press release, but it must mention the embargo date and Nature Structural & Molecular Biology. If you or your Press Office have any enquiries in the meantime, please contact press@nature.com.

Please note that *Nature Structural & Molecular Biology* is a Transformative Journal (TJ). Authors may publish their research with us through the traditional subscription access route or make their paper immediately open access through payment of an article-processing charge (APC). Authors will not be required to make a final decision about access to their article until it has been accepted. Find out more about Transformative Journals <https://www.springernature.com/gp/open-research/transformative-journals>

Authors may need to take specific actions to achieve [compliance with funder and institutional open access mandates](https://www.springernature.com/gp/open-research/funding/policy-compliance-faqs). If your research is supported by a funder that requires immediate open access (e.g. according to [Plan S principles](https://www.springernature.com/gp/open-research/plan-s-compliance)) then you should select the gold OA route, and we will direct you to the compliant route where possible. For authors selecting the subscription publication route, the journal's standard licensing terms will need to be accepted, including [self-archiving policies](https://www.springernature.com/gp/open-research/policies/journal-policies). Those licensing terms will supersede any other terms that the author or any third party may assert apply to any version of the manuscript.

In approximately 10 business days you will receive an email with a link to choose the appropriate publishing options for your paper and our Author Services team will be in

touch regarding any additional information that may be required.

Sincerely,

Carolina Perdigoto, PhD
Chief Editor
Nature Structural & Molecular Biology
orcid.org/0000-0002-5783-7106
